# The chromatin, topological and regulatory properties of pluripotency-associated poised enhancers are conserved in vivo

Giuliano Crispatzu [1,2,3,8✉], Rizwan Rehimi[1,8], Tomas Pachano [1], Tore Bleckwehl[1], Sara Cruz-Molina [4], Cally Xiao [1,3,5,6], Esther Mahabir [1], Hisham Bazzi [1,3,5] & Alvaro Rada-Iglesias [1,3,7✉]

Poised enhancers (PEs) represent a genetically distinct set of distal regulatory elements that control the expression of major developmental genes. Before becoming activated in differentiating cells, PEs are already bookmarked in pluripotent cells with unique chromatin and topological features that could contribute to their privileged regulatory properties. However, since PEs were originally characterized in embryonic stem cells (ESC), it is currently unknown whether PEs are functionally conserved in vivo. Here, we show that the chromatin and 3D structural features of PEs are conserved among mouse pluripotent cells both in vitro and in vivo. We also uncovered that the interactions between PEs and their target genes are globally controlled by the combined action of Polycomb, Trithorax and architectural proteins. Moreover, distal regulatory sequences located close to developmental genes and displaying the typical genetic (i.e. CpG islands) and chromatin (i.e. high accessibility and H3K27me3 levels) features of PEs are commonly found across vertebrates. These putative PEs show high sequence conservation within specific vertebrate clades, with only a few being evolutionary conserved across all vertebrates. Lastly, by genetically disrupting PEs in mouse and chicken embryos, we demonstrate that these regulatory elements play essential roles during the induction of major developmental genes in vivo.

[1] Center for Molecular Medicine Cologne (CMMC), University of Cologne, Cologne, Germany. [2] Department of Internal Medicine II, University Hospital Cologne, Cologne, Germany. [3] Cluster of Excellence for Aging Research (CECAD), University of Cologne, Cologne, Germany. [4] Max Planck Institute for Molecular Biomedicine, Muenster, Germany. [5] Department of Dermatology and Venereology, University Hospital Cologne, Cologne, Germany. [6] Laboratory of Neuro Imaging, USC Stevens Neuroimaging and Informatics Institute, Keck School of Medicine of University of Southern California, Los Angeles, CA, USA. [7] Instituto de Biomedicina y Biotecnología de Cantabria (IBBTEC), CSIC-Universidad de Cantabria-SODERCAN, Santander, Spain. [8]These authors contributed equally: Giuliano Crispatzu, Rizwan Rehimi. ✉email: gcrispat@uni-koeln.de; alvaro.rada@unican.es

Poised enhancers (PEs) were originally described in ESC[1,2] as a rather limited set of distal regulatory elements that, before becoming activated upon cellular differentiation, are already bookmarked in pluripotent cells with unique chromatin and topological features. Briefly, in ESC, PEs are already bound by transcription factors and co-activators (e.g. p300), display high chromatin accessibility and are marked with H3K4me1, which are all typical features of active enhancers (Supplementary Fig. 1A). However, in contrast to active enhancers, PEs are not marked with H3K27ac, but are bound instead by Polycomb Group protein complexes (PcG[3]) and their associated histone modifications (e.g. H3K27me3). Moreover, it was previously reported that PEs can physically interact with their target genes already in ESC[4] in a PcG-dependent manner[5]. Most importantly, PEs were shown to be essential for the proper induction of their target genes upon differentiation of ESC into Anterior Neural Progenitor Cells (AntNPC)[5]. The previous epigenetic and topological features could explain, at least partly, why PEs are essential for the induction of major developmental genes. More recently, we also showed that the unique epigenetic, topological, and regulatory properties of PEs are genetically encoded and critically dependent on the presence of CpG islands (CGIs) that are not associated with annotated promoters and that are referred to as orphan CGI (oCGI)[6–9]. Altogether, this led us to suggest that the genetic, epigenetic, and topological features of PEs could endow developmental loci with a permissive regulatory landscape that facilitates the precise and specific induction of PE–target genes upon pluripotent cell differentiation[5,9]. However, the previous characterization of PEs was based on the analyses of a few loci in which ESC were used as an in vitro differentiation model. Therefore, it is still unclear whether PEs exist and display essential regulatory functions in vivo.

By generating and mining various types of genomic data, here we show that PEs display their characteristic genetic, epigenetic, and topological features in both in vitro and in vivo pluripotent cells. Furthermore, we also show that PEs are pervasively found across vertebrates, although they tend to be preferentially conserved within specific vertebrate clades. Chiefly, by deleting conserved PEs in mouse and chicken embryos, we conclusively demonstrate that this type of regulatory elements is essential for the proper expression of major developmental genes during vertebrate embryogenesis.

## Results

### Poised enhancers (PEs) display their characteristic chromatin signature in the mouse post-implantation epiblast.
Poised, active, and primed enhancers were previously identified in mESC grown under serum+LIF (S + L) conditions[5], which only recapitulates part of the pluripotency states that exist both in vitro and in vivo[10,11]. Therefore, to gather a more complete view of the full repertoire of pluripotency-associated PEs, we analyzed the necessary data (i.e. ATAC-seq, H3K27ac ChIP-seq, H3K27me3 ChIP-seq, H3K4me1 ChIP-seq) to identify these regulatory elements in 2i mESC (naïve pluripotency) and EpiLC (epiblast-like cells; formative pluripotency) ("Methods"; Supplementary Data 1). Next, PEs identified in S + L mESC, 2i mESC, and EpiLC were combined, resulting in a total of 4191 unique mouse PEs (Fig. 1a; Supplementary Fig. 1A–C). A subset of these PEs, which we refer to as PoiAct enhancers ($n = 354$), gets activated in AntNPC as they overlap H3K27ac peaks identified in these cells[5] (Fig. 1a). A similar strategy was used to identify active (high chromatin accessibility/p300 binding, H3K27ac+; $n = 14803$) and primed enhancers (H3K4me1+, H3K27ac−, H3K27me3−; $n = 55812$) in the different mouse in vitro pluripotent cell types ("Methods"; Supplementary Fig. 1A–C). Once these combined

enhancer sets were identified, we investigated their epigenetic profiles during early mouse development (i.e. fertilization (E0) to gastrulation (E6.5)). To this purpose, we mined publically available ATAC-seq and H3K27me3 ChIP-seq data sets[12–14] and also generated relevant data in mouse E6.5 epiblast (i.e. ATAC-seq, H3K27ac ChIP-seq). Regarding H3K27me3 levels at PEs, we found that this histone modification is especially high in sperm[14,15], while it progressively accumulates during oocyte development (Supplementary Fig. 2A). Subsequently, H3K27me3 becomes erased upon fertilization and then it progressively increases until it reaches high levels in the post-implantation epiblast (E5.5–6.5) (Fig. 1b), thus resembling how this histone modification accumulates at bivalent promoters[14]. Similarly, chromatin accessibility (i.e. measured by ATAC-seq) progressively increases at PEs following fertilization, reaching its highest levels in the post-implantation epiblast (E6.5) (Fig. 1b). Therefore, the chromatin signature that PEs display in mESC (i.e. high chromatin accessibility and H3K27me3) is also observed in vivo in pluripotent cells from the post-implantation epiblast. Importantly, the PEs remain inactive in the post-implantation epiblast, as they display low H3K27ac (Fig. 1c). In contrast, and in agreement with our previous observations, the PEs became active and, thus, gained H3K27ac at later developmental stages (e.g. E12.5), particularly in the developing brain (Supplementary Fig. 2B). To further illustrate the existence of PEs in vivo, we used ATAC-seq and ChIP-seq data generated in the mouse E6.5 epiblast ("Methods") to directly call PEs in vivo (Fig. 1c). Out of the 3057 PEs identified in the E6.5 epiblast, 39.68% (1213/3057) overlapped with the PEs identified in the in vitro pluripotent cells and 41.09% (1256/3057) became activated in the E10.5 brain (i.e. in vivo PoiAct enhancers). On the other hand, 30.06% (1260/4191) of the in vitro PE were also called in vivo. Considering the current limitations to generate high-quality ChIP-seq data in the mouse epiblast, these results further support the presence of PEs in pluripotent cells both in vitro and in vivo.

### Mouse PEs display high genetic and epigenetic conservation across mammals.
Having confirmed that PEs display their characteristic chromatin features in vivo, we then assessed their evolutionary conservation among vertebrate genomes ("Methods"; Fig. 1d; Supplementary Fig. 2C, D), as this can provide preliminary insights into the functional relevance of non-coding sequences[16]. In general, mouse PEs display high sequence conservation across mammals, which then decreases in non-mammalian amniotes (i.e. birds and reptiles) and is already quite moderate in anamniotes (i.e. amphibians and fish) (Fig. 1d; Supplementary Fig. 2C, D). Furthermore, PEs are considerably more conserved than active enhancers across all vertebrates (FC = 1.86, $p = 2.98e−06$; two-sided Wilcoxon test), probably reflecting the potential involvement of PEs in highly conserved developmental processes (e.g. organogenesis, patterning)[1,5]. Notably, despite their high evolutionary conservation, PEs display a limited overlap with "ultraconserved non-regulatory elements" (uCNEs)[17] (Supplementary Fig. 2E), which, at least in some cases, act as enhancers of important developmental genes[18,19]. This suggests that, despite their overall high sequence conservation, PEs and uCNEs represent distinct classes of distal regulatory elements.

Having shown that mouse PEs display high genetic conservation, we then wanted to evaluate whether their unique chromatin signature (i.e. high chromatin accessibility/p300 binding, high H3K27me3/PcG binding) was evolutionary conserved. To this end, we mined ATAC-seq and ChIP-seq data previously generated in ESC from different mammals (human[1], chimp[20]), as well as in early stages of zebrafish development[21]. In addition,

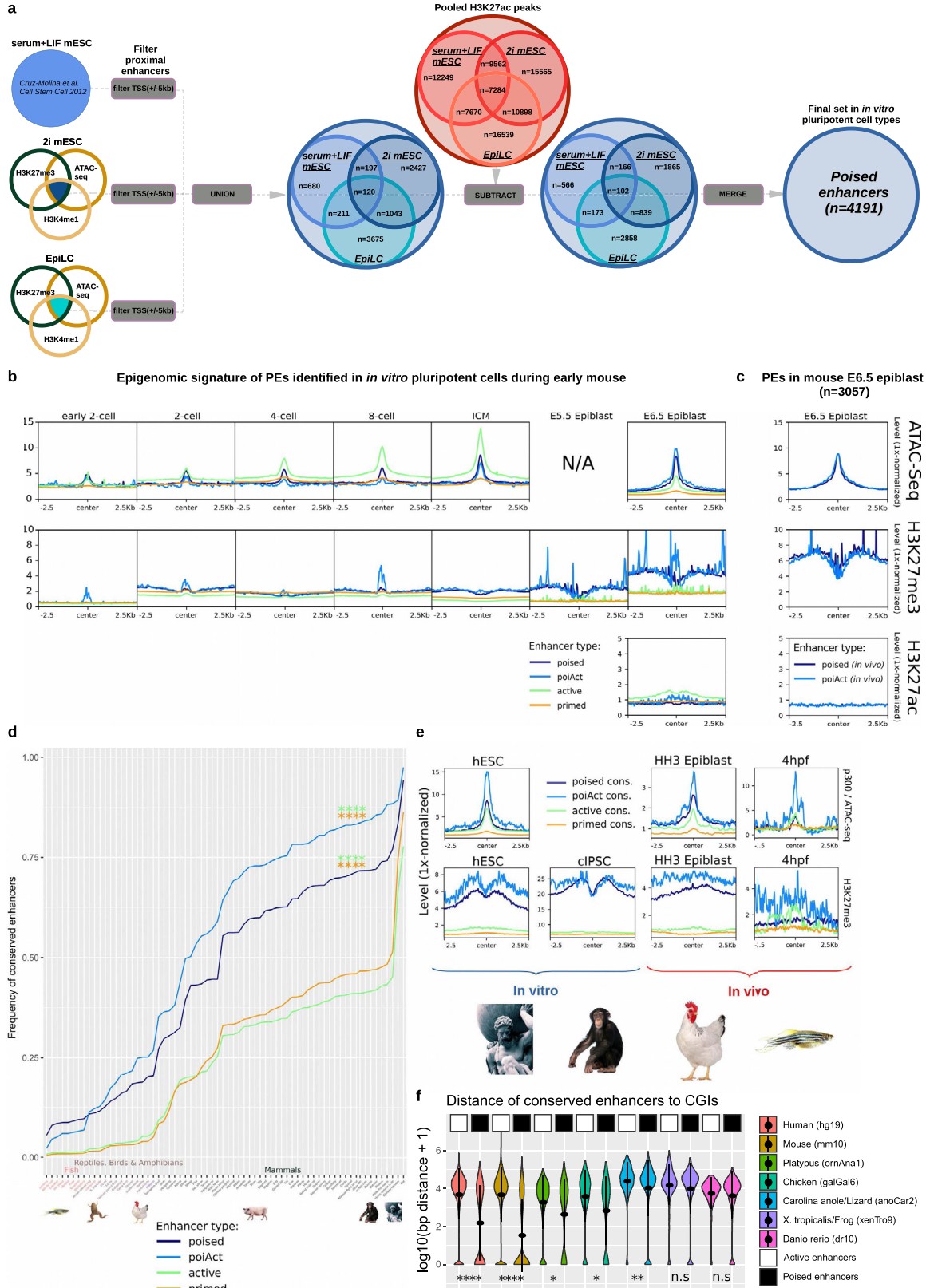

**b** Epigenomic signature of PEs identified in *in vitro* pluripotent cells during early mouse

**c** PEs in mouse E6.5 epiblast (n=3057)

Enhancer type:
- poised
- poiAct
- active
- primed

Enhancer type:
- poised *(in vivo)*
- poiAct *(in vivo)*

**e** In vitro / In vivo

poised cons.
poiAct cons.
active cons.
primed cons.

**f** Distance of conserved enhancers to CGIs

- Human (hg19)
- Mouse (mm10)
- Platypus (ornAna1)
- Chicken (galGal6)
- Carolina anole/Lizard (anoCar2)
- X. tropicalis/Frog (xenTro9)
- Danio rerio (dr10)
- Active enhancers
- Poised enhancers

Enhancer type:
- poised
- poiAct
- active
- primed

we also generated ATAC-seq and H3K27me3 ChIP-seq data for the chicken epiblast (HH3). Next, we analyzed the chromatin features of the mouse PEs that were conserved in each of the previous vertebrate species (Fig. 1e). In mammalian ESC and to a lesser extent in the chicken epiblast, PEs showed the expected chromatin signature (i.e. high H3K27me3 and ATAC-seq levels),

while this was less obvious in zebrafish embryos, especially for H3K27me3. There are probably several reasons contributing to these differences: (i) the conserved PEs in non-mammalian vertebrates might include TF binding sites (indirectly detected by p300/ATAC peaks) but not nearby oCGI, which are responsible for PcG recruitment and H3K27me3 enrichment[9] (see below); (ii)

**Fig. 1 Mouse poised enhancers (PEs) display their characteristic chromatin signature in vivo and are highly conserved in mammals. a** ATAC-seq and ChIP-seq data for p300, H3K4me1, H3K27me3, and H3K27ac were used to call PEs in three distinct in vitro pluripotent states (i.e. 2i ESC, S + L ESC and EpiLC)[5,94]. PEs are characterized by high p300/ATAC signals, high H3K27me3, and H3K4me1 levels, as well as low H3K27ac levels and located at least 5 kb apart from a TSS ("Methods"). **b** ATAC-seq, H3K27ac, and H3K27me3 signals during early mouse embryogenesis[12–14] (2-cell stage to E6.5 epiblast) are shown around poised, active, primed and PoiAct enhancers identified in in vitro pluripotent cells. **c** Public available H3K27me3[14], as well as generated ATAC-seq and H3K27ac signals during early mouse embryogenesis (E6.5 epiblast) are shown around poised and PoiAct enhancers identified in in vivo. **d** Sequence conservation across 68 vertebrate species (Supplementary Data 1) was measured for poised, active, primed, and PoiAct enhancers using a mappability threshold of 0.5. PEs and PoiAct enhancers show significantly higher conservation than active (FC = 1.86, $p$ = 2.98e−06; FC = 2.2, $p$ = 5.25e −08 respectively; two-sided Wilcoxon test) or primed enhancers (FC = 1.72, $p$ = 4.44e−06; FC = 2.03, $p$ = 9.61e−08 respectively; two-sided Wilcoxon test). **e** ATAC-seq and H3K27me3 signals from human ESC (hESC), chimpanzee iPSC (cIPSC), epiblast from HH3 chicken embryos and 4 hpf zebrafish embryos are shown around poised, active, primed and PoiAct enhancers identified in mouse pluripotent cells and conserved in each of the corresponding vertebrate species. **f** Distance between active or poised mouse enhancers conserved in the indicated vertebrate species and CGIs identified in the same species using Bio-CAP[6, 22]. The asterisks indicate that PEs are significantly closer to CGIs than active enhancers in all species except the frog (two-sided Wilcoxon test). The exact overlap of PEs (compared to active enhancers) with CGIs increases in all these species: in human from 9.85% to 43.14% ($p$ = 1.18e−276), in mouse from 10.88% to 58.91% ($p$ < 2.2e−16), in platypus from 13.09% to 29.68% ($p$ = 2.51e−06), in chicken from 9.93% to 25.95% ($p$ = 1.37e−14), in lizard from 3.23% to 10.56% ($p$ = 0.017), in *Xenopus tropicalis* from 3.90% to 8.72% ($p$ = 0.94), and in zebrafish from 2.99% to 9.07% ($p$ = 0.23). *$p$ ≤ 0.05, **$p$ ≤ 0.01, ***$p$ ≤ 0.001, ****$p$ ≤ 0.0001.

the lower number of conserved PEs in non-mammalian vertebrates (chicken $n$ = 767, zebrafish $n$ = 354) can make the ATAC-seq and ChIP-seq profiles to look noisier; (iii) the data for non-mammalian vertebrates was generated in vivo, which due to lower cell numbers and higher cellular heterogeneity can also compromise the quality of the ATAC-seq and ChIP-seq profiles. Nevertheless, even in non-mammalian vertebrates (i.e. chicken and zebrafish), the conserved PEs display ATAC-seq and H3K27me3 signals clearly higher than those observed for other enhancer classes.

**PEs are a prevalent feature of vertebrate genomes**. We previously found that PEs have a unique and modular genetic composition[5,9], since compared to other enhancer classes, they are frequently located close to oCGI (i.e. 70–80% of PEs are located within 3 kb of computationally defined "weak" CGIs[5] or biochemically defined CGIs[9], also known as Non-Methylated Islands (NMI)[22]). Our previous work based on the in-depth analyses of a few selected PE loci suggests that the proximity to CGI confers PEs with unique epigenetic features, such as binding by PcG and DNA hypomethylation[5,9,23–25]. Congruently, analysis of whole-genome bisulfite sequencing data generated in mouse pluripotent cells[26,27] revealed that PEs are globally hypomethylated both in vitro and in vivo (Supplementary Fig. 3A). Moreover, PEs are bound in mESC by KDM2B (Supplementary Fig. 3B), a protein containing CXXC domains that specifically recognize CGI and that might be responsible, at least partly, for the unique epigenetic properties of the PEs[24,28,29] (Supplementary Fig. 3C). Given the importance that CGI might have in conferring PEs with their unique chromatin and regulatory features, we then wanted to evaluate whether the proximity between PEs and CGI was evolutionary conserved. However, CGIs have been traditionally identified using algorithms originally implemented in mammalian genomes, but that, due to the variability in overall GC and CpG contents, do not perform well when applied to cold-blooded vertebrate genomes. To overcome these limitations, we used data previously generated in seven different vertebrates by Bio-CAP (biotinylated CxxC-affinity purification[6,22]), an assay that enables the unbiased identification of CGI. The CGIs identified through Bio-CAP are typically referred to as NMI[22]. However, to avoid possible confusions, from now on we will simply use the term CGI regardless of whether these genetic features were identified based on their genetic composition or by Bio-CAP. Next, we measured the distance between those mouse PEs conserved in each vertebrate species and the nearest CGI (Fig. 1f; Supplementary Fig. 3D). Among the seven vertebrate

species, the proximity of PEs to CGIs was particularly obvious in mammals (mouse, human, platypus) and, to a lesser extent, in chicken. Similarly, we observed that conserved PEs tended to be considerably closer to CGIs than their active counterparts in both mammals and chicken, but not in anamniotes (i.e. frog and zebrafish) (Fig. 1f). Therefore, considering the important role of oCGI in mediating the recruitment of PcG to PEs in mESC[9], the weaker H3K27me3 enrichment observed at conserved PEs in zebrafish embryos (Fig. 1e) might be explained, at least partly, by the frequent absence of nearby oCGI.

Together with our sequence conservation analyses, the previous results suggest that PEs are prevalent in mammals but rather scarce in other vertebrates, especially in anamniotes. Alternatively, PEs might be abundant in all vertebrates, but sequence conservation, including the proximity to oCGI, might preferentially occur within individual vertebrate clades. To distinguish between these two possibilities, we called PEs de novo in human ESC, chicken epiblast and zebrafish embryos using available epigenomic data sets[1,21,30] (Fig. 2a) and similar criteria to the ones used to identify PEs in mouse cells (Fig. 1a). The term de novo is used to define PEs that are directly identified using epigenomic data generated in each of the investigated vertebrate species in contrast to those solely defined by sequence conservation (Fig. 1d–f). Notably, these de novo PEs were abundant in both mammals (human $n$ = 4009) and non-mammals (chicken $n$ = 7306, zebrafish $n$ = 2534) and, especially in zebrafish, they displayed stronger ATAC-seq and H3K27me3 signals than PEs identified through conservation analyses (Fig. 2a; Fig. 1e). Accordingly, while in human ESC and chicken epiblast both de novo and conserved PEs were similarly close to CGIs, in the zebrafish embryos the proximity to CGIs was dramatically increased for the de novo PEs (Fig. 2b). Furthermore, in all the investigated species, the de novo PE was strongly associated with genes involved in developmental processes, such as patterning and organogenesis (Fig. 2c; Supplementary Fig. 3E, F). Overall, these results indicate that PEs showing similar genetic (i.e. proximity to oCGI) and epigenetic (i.e. high ATAC/p300 and H3K27me3 levels) features are prevalent in all vertebrates, but that a large fraction of them might be specific to each vertebrate class/group. In agreement with this possibility, chicken and zebrafish de novo PEs were highly conserved among birds/reptiles and fish, respectively, but not in other vertebrates (Fig. 2d), thus resembling the preferential conservation of mouse PE within mammals (Fig. 1d). Nevertheless, we also noticed that those de novo PE showing high sequence conservation tended to be even closer to CGIs (Supplementary Fig. 3G) and, displayed higher

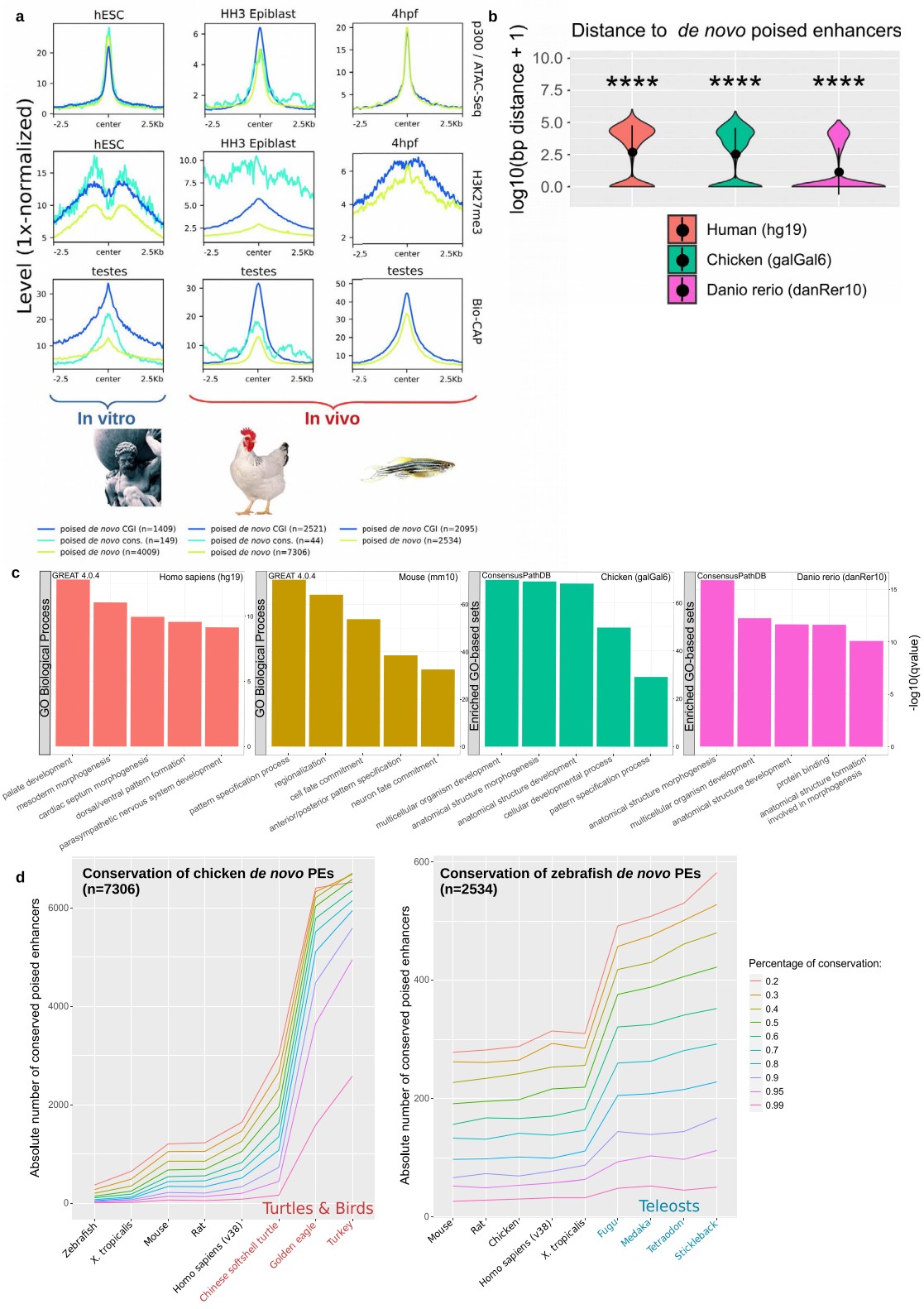

H3K27me3 levels (Fig. 2a), which according to our recent work[9], might endow PEs with particularly privileged regulatory properties.

**PEs globally interact with bivalent gene promoters in pluripotent cells both in vitro and in vivo.** The presence of CGI in the vicinity of PEs provides them with unique topological properties. Namely, PEs can already physically interact with their target genes in ESC, thus preceding their activation in neural progenitors[5,9]. This might confer anterior neural loci with a permissive regulatory topology that facilitates the precise and robust induction of PE–target genes. However, this model is based on the detailed analysis of a few endogenous (i.e. *Sox1, Six3, Lmx1b, Lhx5*) or ectopic (i.e. *Gata6*) PE loci using 4C-seq

**Fig. 2 PEs are a widespread feature across vertebrates. a** PEs were called de novo using available ATAC-Seq, p300 ChIP-seq and H3K27me3 ChIP-Seq data generated in human embryonic stem cells (hESC), HH3 chicken epiblast and zebrafish 4hpf zebrafish embryos (sphere stage; blastulation). ATAC-seq, ChIP-seq, and Bio-CAP[6,22] signals are shown for each species around all de novo PEs as well as those overlapping CGIs or previously called conserved mouse PEs (except for zebrafish due to low numbers; $n = 2$). **b** Distance between de novo PEs and CGIs for each of the investigated vertebrate species. The asterisks indicate that de novo PEs are significantly closer to CGIs than active mouse enhancers conserved in each species. The exact overlap of de novo PEs (compared to active enhancers) with CGIs increases in all species: in human from 9.85% to 33.18% ($p = 6.64e−69$; two-sided Wilcoxon test); in chicken from 9.93% to 34.15% ($p = 9.88e−27$; two-sided Wilcoxon test) and in zebrafish from 2.99% to 68.77% ($p = 5.57e−61$; two-sided Wilcoxon test). **c** De novo PE sets were annotated using GREAT 4.0.4[97] for human and mouse. As this software is not available for chicken and recent zebrafish versions, these de novo PEs were linked to the nearest gene and the resulting gene set was analyzed using ConsensusPathDB[98]. **d** The sequence conservation of de novo PEs identified in chicken (left panel) and zebrafish (right panel) was measured across representative species of the main vertebrate clades using different mappability thresholds (mapping ratios 0.2–0.99; color legend). $*p \leq 0.05$, $**p \leq 0.01$, $***p \leq 0.001$, $****p \leq 0.0001$.

technology[5,9]. In principle, the generality of our previous observations could be addressed using Hi-C technology. However, to detect enhancer-promoter contacts at high resolution, Hi-C typically requires large and cost-prohibitive sequencing depths[31,32]. Therefore, we decided to use a more targeted approach, called HiChIP[33], that combines Hi-C and ChIP, thus allowing interrogation of DNA loops associated with proteins/histone marks of interest. As both PEs and their target gene promoters are typically enriched in H3K27me3[1,5], we first generated HiChIPs for H3K27me3 in mESC grown under serum +LIF conditions as biological duplicates. Both replicates were pooled to call loops ($p < 0.01$; $n = 72265$; ranging from 25 kb–1.81 Mb in loop size; "Methods"). In order to validate the quality of the previous HiChIP data and its usefulness to identify PE–target gene interactions, we first evaluated the four PE loci previously analyzed by 4C-seq and confirmed the reported contacts (Fig. 3a; Supplementary Fig. 4A). Moreover, when considering all the detected HiChIP loops, we found that a large fraction of distal PEs ($n = 3239$; >10 kb from a TSS) interacts with at least one locus (55.7%). A large fraction of these interactions (39.8%) were with gene promoters (Fig. 3b), which, importantly, often display a bivalent state in mESC (Fig. 3c). More specifically, the chromatin state of 1083 out of the 1585 TSSs interacting with distal PEs in mESC (Fig. 3c) has been previously described[34] and, among them, 424 are bivalent ($n = 424/2794$; $p < 2.2e−16$; OR = 2.63; Fisher test), 40 are only marked with H3K27me3 ($n = 40/160$; $p = 8.88e−14$; OR = 4.90; Fisher test), 480 are only marked with H3K4me3 ($n = 480/9663$; $p = 2.64e−06$; OR = 0.77; Fisher test) and 155 are unmarked ($n = 155/4758$; $p < 2.2e−16$; OR = 0.495; Fisher test). In agreement with this significant over-representation of bivalent and H3K27me3-only genes, the genes interacting with PEs tend to be preferentially involved in developmental processes (e.g. patterning, morphogenesis) (Fig. 3d). Interestingly, we noticed that in contrast to the frequent long-range/inter-TAD interactions established between PcG-bound genes/domains[3,4,35], PE–gene contacts preferentially occur within the same TAD (i.e. intra-TAD) (Fig. 3e). Moreover, the interactions between PEs and bivalent genes detected by HiChIP ($n = 526$) were also readily observed in Hi-C data generated in mESC grown under both serum+LIF and 2i conditions[36] (Fig. 3f). This further validates the quality of our HiChIP data and supports that PE–gene contacts are present across different in vitro pluripotent states (Supplementary Fig. 1A). Once these pre-formed contacts between PEs and their target genes in mESC were globally confirmed, we also investigated whether this topological feature could be also observed in vivo using Hi-C data recently generated in peri-implantation mouse embryos (E3.5–E7.5)[37]. Notably, we observed clear contacts between PEs and bivalent promoters in the mouse ICM as well as in the post-implantation epiblast (Fig. 3g). This is in agreement with the overall conservation of the PEs chromatin features in vivo (Fig. 1b, c), since such features,

namely H3K27me3/PcG, are considered as important mediators of PE–gene interactions[5].

**The physical communication between PEs and their target genes depends on the combined action of Polycomb, Trithorax, and architectural proteins.** There is some discrepancy regarding which PcG complexes, PRC1 or PRC2, contribute to the physical communication between PcG-bound loci, including PE–target gene interactions[4,5,38–40]. To investigate whether PRC1 and/or PRC2 are globally involved in the establishment of PE–target gene interactions in mESC we used PRC2 (EED$^{−/−}$ mESC[41]) and PRC1 (RING1a$^{−/−}$RING1b$^{fl/fl}$ mESC[42] treated with Tamoxifen for 72 h) null mESC lines (Supplementary Fig. 4B). In contrast to H3K27me3, which is globally lost in PRC1 and PRC2 null mESC[5,43], H3K4me3 levels are maintained or even increased at gene promoters in general and bivalent ones in particular[44] (Supplementary Fig. 4C), thus enabling the use of this histone mark to study PE–gene interactions in PcG-null ESC. Therefore, we generated H3K4me3 HiChIP data as biological duplicates in each PcG-null mESC line. Overall, we observed that PRC1 null ESC showed a reduction in mid-range interactions and in the overall number of loops, probably reflecting the involvement of PRC1 in active enhancer-gene contacts[40] (Fig. 4a). Most importantly, the contacts between PEs and bivalent promoters were globally reduced in PRC1 null cells (Fig. 4b). In contrast, in PRC2 null cells we observed strongly reduced PE–gene interactions within previously investigated loci[5] (Fig. 4c; Supplementary Fig. 4D), but only mild effects at a global level (Fig. 4b). The importance of PRC1 for PE–gene interactions was also confirmed using Hi-C data generated in different RING1A/B-depleted mESC lines (RING1a$^{−/−}$RING1b$^{DEG}$ mESC) (Fig. 4d). Therefore, our global analyses indicate that PRC1 is required for proper PE–target gene communication in ESC. This is likely to be mediated by canonical PRC1 (cPRC1) through the polymerization and/or phase-separation capacity of its PHC and CBX subunits, respectively[34,45–49].

It has been recently shown that the long-range interactions between PcG-bound genes/domains in mouse pluripotent cells are controlled not only by PRC1 but also by other protein complexes (i.e. Trithorax/MLL2, Cohesin)[50]. Using Hi-C data generated in E6.5 epiblasts from *Kmt2b*$^{ko}$ (*Mll2*$^{ko}$) mice[51], we observed that the contacts between PEs and bivalent promoters were also reduced in the *Kmt2b* mutant epiblasts (Fig. 4e). We then analyzed Hi-C data generated in a Scc1/Rad21-Degron ESC line[52] and found that, while Cohesin depletion led, as previously reported[52], to increased long-range interactions between PcG domains, it actually reduced intra-TAD contacts between PE and their bivalent target genes (Fig. 4d). Similarly, CTCF depletion[53] also diminished the interactions between PE and bivalent genes (Fig. 4f). Therefore, PE-bivalent gene contacts might be the result of the combined action of two major mechanisms; (i) homotypic chromatin

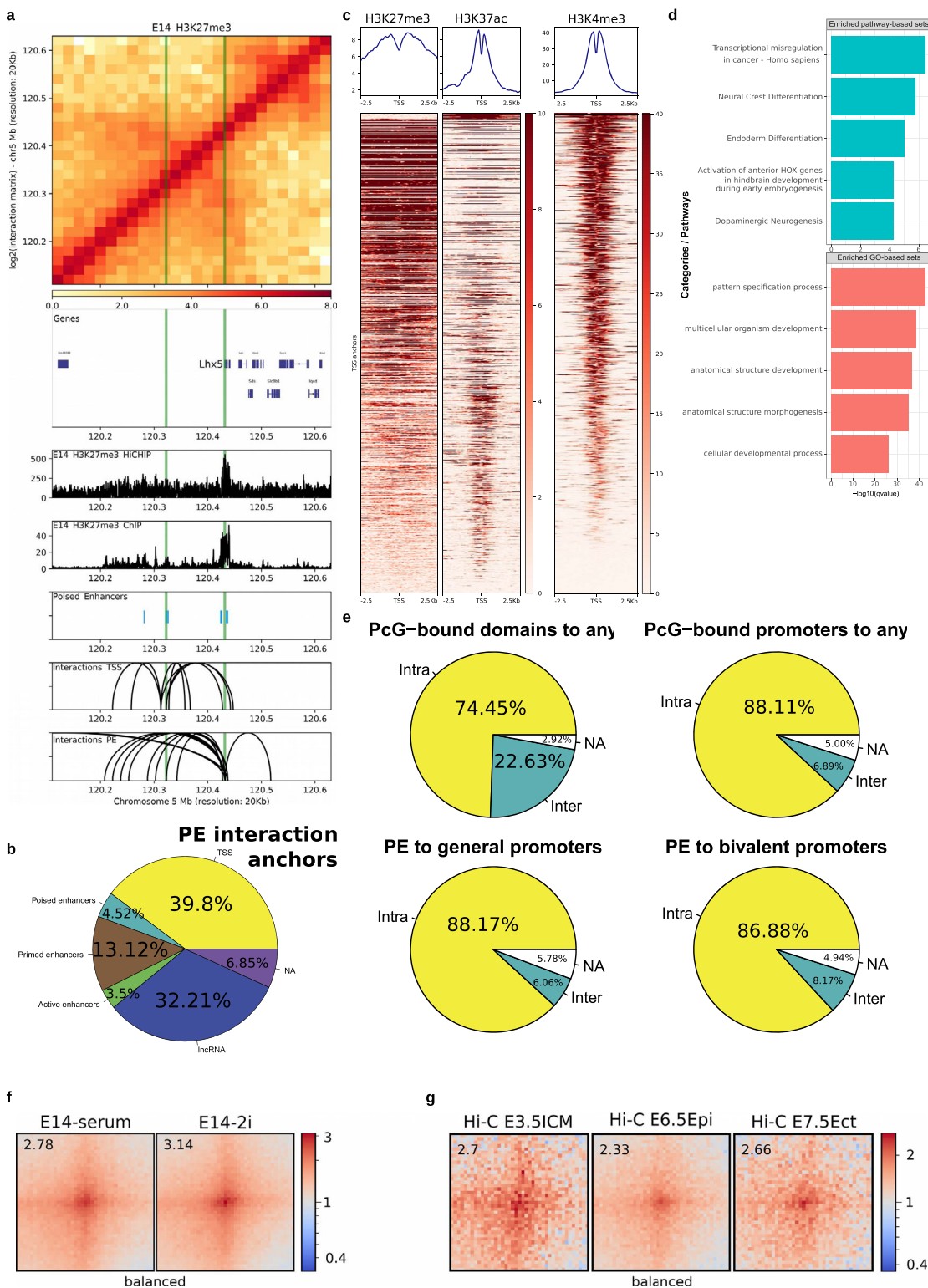

interactions mediated by PcG and Trithorax complexes and (ii) loop extrusion mechanisms dependent on Cohesin and CTCF that favor PE–gene communication within TADs (Fig. 3e).

**The physical interactions between PEs and their target genes are maintained once PE get activated.** Our previous analyses based on 4C-seq experiments indicated that PE–gene contacts

already present in ESC are maintained in AntNPC once the PE and their target genes become active. However, due to the cellular heterogeneity present within AntNPC, PEs and their target genes get activated only in a fraction of cells that cannot be specifically interrogated by 4C-seq/Hi-C experiments. Therefore, we conducted H3K27ac HiChIP experiments in E14 AntNPC[54] and E10.5 mouse brains to specifically and globally evaluate

**Fig. 3 Mouse PEs globally interact with bivalent promoters in pluripotent cells both in vitro and in vivo. A** H3K27me3 HiChIP and ChIP-seq profiles generated in mESC are shown around the *Lhx5* locus. Both the *Lhx5* TSS and a PE previously shown to control this gene in AntNPC (PE *Lhx5* (−109kb)) are marked with green vertical lines. Upper panel: heatmap of log2 interaction intensities based on H3K27me3 HiChIP data generated in mESC. Two medium panels: ChIP-seq and 1D HiChIP signals for H3K27me3 in mESC. Lower two panels: significant ($p < 0.01$; FitHiChIP[102]) interactions were called using the H3K27me3 HiChIP PETs (paired-end tags) and loops in which one of the anchors overlaps (±10 kb) either the PE *Lhx5* (−109kb) or the *Lhx5* TSS are shown. **b** Significant ($p < 0.01$; FitHiChIP) interactions were called using the mESC H3K27me3 HiChIP PETs (paired-end tags). Interaction anchors were overlapped with PEs and their interaction partners were hierarchically annotated with the categories shown in the pie chart ("Methods"). **c** mESC ChIP-seq profiles for H3K27me3, H3K27ac, and H3K4me3 are shown around the TSS ($n = 1083$) interacting with PEs according to the H3K27me3 HiChIP loops identified in mESC ($p < 0.01$; FitHiChIP). **d** Pathway (green) and Gene Ontology (red) analysis for the TSS/genes interacting with PEs according to the H3K27me3 HiChIP loops identified in mESC ($p < 0.01$; FitHiChIP) was performed with ConsensusPathDB. **e** The interactions (called using mESC H3K27me3 HiChIP data; $p < 0.01$; FitHiChIP) established by PcG domains ("Methods") and PcG-bound promoters as well as those involving PE-general promoter or PE-bivalent promoter pairs were classified as either intra-TAD (yellow) or inter-TAD (green) depending on whether the two anchors of each loop occur within the same or different TADs, respectively. **f, g**) Significant (mESC H3K27me3 HiChIP data; $p < 0.01$; FitHiChIP) interactions between distal PEs (>10 kb from TSS) and bivalent promoters ($n = 526$) were visualized as pileup plots using Hi-C data generated in vitro[36] (2i ESC, serum+LIF ESC; **f**) and in vivo[37] (E3.5 Inner Cell Mass (ICM), E6.5 epiblast, E7.5 ectoderm; **g**). Hi-C pairwise interactions are shown 250 kb up- and downstream of each PE and bivalent promoter pair. Hi-C interaction matrices were KR-balanced (balanced). The numbers in the upper left corners correspond to "loopiness" values (center intensity normalized to the intensity in the corners).

interactions established by PE that became active (i.e. PoiAct enhancers) in these cells. First, we evaluated the PE loci previously analyzed by 4C-seq and confirmed that PE–target gene contacts are maintained once PE becomes active in both AntNPC and E10.5 brain cells (Fig. 5a, b; Supplementary Fig. 5A). More importantly, we found that distal PE-bivalent gene contacts detected in mESC (Fig. 3b) were also frequently observed once PEs became active in the developing mouse brain (50.76–54.56% overlap; $n = 67,748–94,731$ loops; $p < 0.05$; ±10 kb anchor extension) (Fig. 5c) or in AntNPC (36.69% overlap; $n = 49,552$ loops; $p < 0.05$; ±10 kb anchor extension) (Fig. 5d). Moreover, distal PoiAct enhancer contacts were also found in AntNPC (55.90% interacting with at least one locus) and the E10.5 brain (59.07–65.95% interacting with at least one locus), while their target genes were induced in their respective differentiating and developmental stages (Supplementary Fig. 5B–D; Supplementary Fig. 6A, B).

**Highly conserved PEs are necessary for the induction of major brain developmental genes in vertebrate embryos.** The previous analyses show that the main genetic (e.g. proximity to CGI), chromatin (e.g. high H3K27me3/PcG levels), and topological (e.g. pre-formed contacts with target genes) features of PEs are evolutionary conserved and detectable in vivo. These observations support, albeit in a correlative manner, the functional relevance of PEs during early vertebrate embryogenesis, particularly during brain development. The functional relevance of PEs was previously demonstrated by deleting candidate PEs in mESC (i.e. PE *Lhx5*(−109 kb), PE *Six3*(−133 kb), PE *Sox1*(+35 kb), PE *Wnt8b* (+21 kb)), which severely compromised the induction of major brain developmental genes (i.e. *Lhx5*, *Six3*, *Sox1*, *Wnt8b*) upon differentiation of ESC into AntNPCs[5]. However, it is currently unknown whether PEs also have essential and non-redundant regulatory functions in vivo or whether, alternatively, these privileged regulatory properties might represent an in vitro "artifact" due to the reduced robustness of in vitro differentiation systems. To start addressing this important question, we first used CRISPR/Cas9 technology to generate mouse embryos in which we deleted the PE *Lhx5*(−109 kb), one of the PEs that we previously characterized in vitro (Fig. 6a–c; Supplementary Fig. 7A, B). Remarkably, the expression of *Lhx5* was strongly reduced in the forebrain of E8.5 and E9.5 PE *Lhx5*(−109 kb)$^{−/−}$ mouse embryos in comparison to their WT isogenic controls (Fig. 6c). Next, since the mouse PE *Lhx5*(−109 kb)$^{−/−}$ has a high genetic and epigenetic conservation across vertebrates, we decided to generate targeted deletions of its homologous sequence in the

developing brain of chicken embryos using CRISPR/Cas9[55] (Fig. 6c–e). Briefly, the forebrain of HH9 chicken embryos was unilaterally electroporated with vectors expressing Cas9 and gRNAs flanking the PE *Lhx5* conserved sequence (Fig. 6d; Supplementary Fig. 7A, C). Subsequently, the expression of *Lhx5* was evaluated by in situ hybridization (ISH) in HH14 chicken embryos (~to E11.5 in mice). Notably, the forebrain expression of *Lhx5* was strongly and specifically reduced in the electroporated side (Fig. 6f). Furthermore, the specificity of these results was further supported by experiments in which the electroporation of chicken embryos with Cas9 and scrambled gRNAs did not affect *Lhx5* expression (Fig. 6f). To further evaluate the in vivo functional relevance of evolutionary conserved PEs, we similarly disrupted another two PEs (PE *Six3*(−133 kb), PE *Sox1*(+35 kb)) that were previously characterized in mESC[5] and that were also genetically and epigenetically conserved in the chicken genome (Supplementary Fig. 7C–E). We again observed a strong reduction in the expression of *Six3* and *Sox1* in the electroporated side of embryos targeted with both Cas9 and gRNAs flanking the PEs, but not when Cas9 was electroporated with scrambled gRNAs (Fig. 6f). Furthermore, in the case of the PE *Six3*(−133 kb) chicken homolog, its disruption resulted in a smaller and malformed eye, in agreement with the strong expression and conserved function of *Six3* during eye development[56] (Fig. 6f). Overall, these results demonstrate that the regulatory function of PEs is essential and conserved in vivo. However, whether the essential regulatory properties of these enhancers require a "poised" state previous to their full activation remains to be demonstrated.

## Discussion

PEs were originally identified, mechanistically dissected, and functionally characterized in ESC[1,5]. Our previous work suggested that PEs could play essential roles during the induction of major developmental genes once pluripotent cells start differentiating[5]. However, it was still unclear whether PEs actually existed and were functionally relevant in vivo. Here we addressed this important question by mining various genomic and epigenomic datasets, which enabled us to conclusively show that PEs not only display their characteristic chromatin signature in vivo, but also that they are a prevalent feature of vertebrate genomes. Interestingly, we found that PEs tend to be highly conserved within specific vertebrate groups (e.g. mouse PEs are highly conserved across mammals; chicken PEs are highly conserved in birds and reptiles), while only a relatively small subset of PEs is conserved across all vertebrates. Nevertheless, in all the

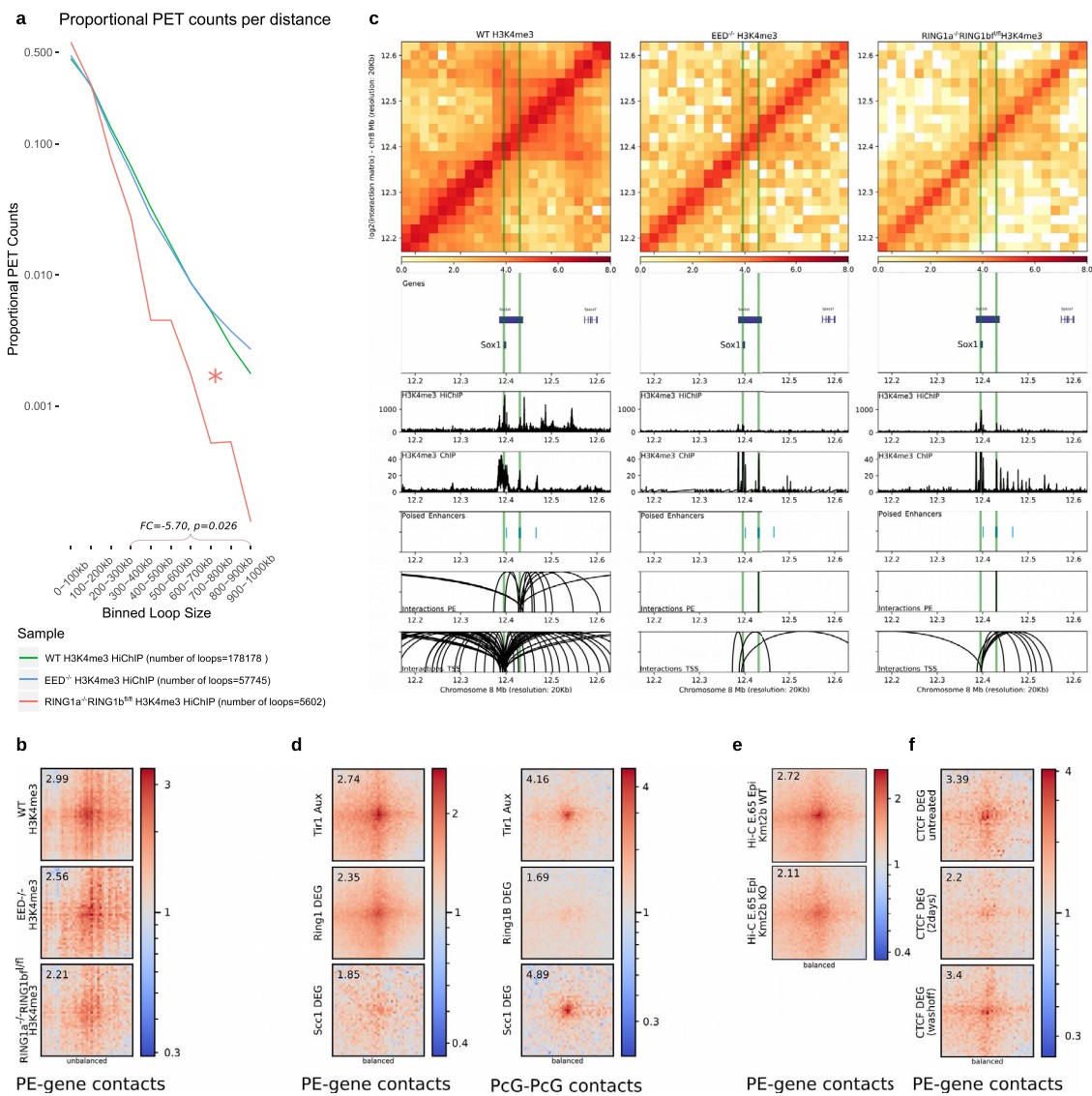

**Fig. 4 Protein complexes controlling the interactions between PEs and bivalent promoters in mESC. a** Significant (q < 0.1; FitHiChIP) H3K4me3 HiChIP interactions identified in WT mESC, $Eed^{-/-}$ mESC and Tamoxifen-treated $Ring1a^{-/-}Ring1b^{fl/fl}$ mESC were plotted according to their loop size. Mid-range (300kb-1mb) loops are significantly reduced (FC = −5.70, p = 0.026; two-sided Wilcoxon test) in Tamoxifen-treated $Ring1a^{-/-}Ring1b^{fl/fl}$ mESC compared to WT. PET = paired-end tags. **b** Significant (mESC H3K27me3 HiChIP data; p < 0.01; FitHiChIP) interactions between distal PEs (>10 kb from TSS) and bivalent promoters (n = 526) were visualized as pileup plots using H3K4me3 HiChIP data generated in WT, $Eed^{-/-}$ and Tamoxifen-treated $Ring1a^{-/-}Ring1b^{fl/fl}$ mESC. HiChIP interaction matrices were coverage-normalized (unbalanced). **c** H3K4me3 HiChIP in WT, $Eed^{-/-}$ mESC, and Tamoxifen-treated $Ring1a^{-/-}Ring1b^{fl/fl}$ mESC around the *Sox1* locus. **d** Significant (mESC H3K27me3 HiChIP data; p < 0.01; FitHiChIP) interactions between distal PEs (>10 kb from TSS) and bivalent promoters (left panel; n = 526) or between PcG domains (right panel; n = 411) were visualized as pileup plots using Hi-C data generated in the following mESC lines expressing *Tir1* and treated with Auxin[52]: WT mESC (*Tir1* Aux), $Ring1a^{-/-}Ring1b^{DEG/DEG}$ mESC (i.e. RING1A/B-depleted cells; Ring1b Aux) and Rad21/Scc1$^{DEG/DEG}$ mESC (i.e. Cohesin depleted cells; Scc1 Aux). Hi-C interaction matrices were KR-balanced (balanced). **e** Significant (mESC H3K27me3 HiChIP data; p < 0.01; FitHiChIP) interactions between distal PEs (>10 kb from TSS) and bivalent promoters (n = 526) were visualized as pileup plots using Hi-C data generated in the epiblast from WT and $Kmt2b/Mll2^{-/-}$ E6.5 mouse embryos[51]. Hi-C interaction matrices were KR-balanced (balanced). **f** Significant (mESC H3K27me3 HiChIP data; p < 0.01; FitHiChIP) interactions between distal PEs (>10 kb from TSS) and bivalent promoters (n = 526) were visualized as pileup plots using Hi-C data generated in CTCF-Degron mESC lines[53] that were either untreated (CTCF untreated), treated with Auxin for two days (CTCF Aux (2 days)) or treated with Auxin and then washed off (CTCF Aux (washoff)). Interaction matrices were KR-balanced (balanced). In (**b**, **d**–**f**), HiChIP or Hi-C pairwise interactions are shown 250 kb up- and downstream of each PE and bivalent promoter pair. The numbers in the upper left corners correspond to "loopiness" values (center intensity normalized to the intensity in the corners).

investigated vertebrate species, PEs were located close to CGIs[6,22] and linked to major developmental genes. We recently showed that orphan CGIs are an essential component of PEs that, together with TAD boundaries, enable them to precisely and specifically control the expression of developmental genes with CpG-rich promoters[9]. We also showed that the main regulatory function of these orphan CGI is to serve as tethering elements that bring PEs and their CpG-rich target genes into physical proximity[9]. Furthermore, the oCGI might also contribute to the high sequence conservation of PEs by protecting them from CpG methylation[9] and, thus, from accumulating C > T mutations. Therefore, we propose that the association of distal enhancers with CGI might represent an ancestral regulatory mechanism in vertebrate genomes that enables the precise and specific induction

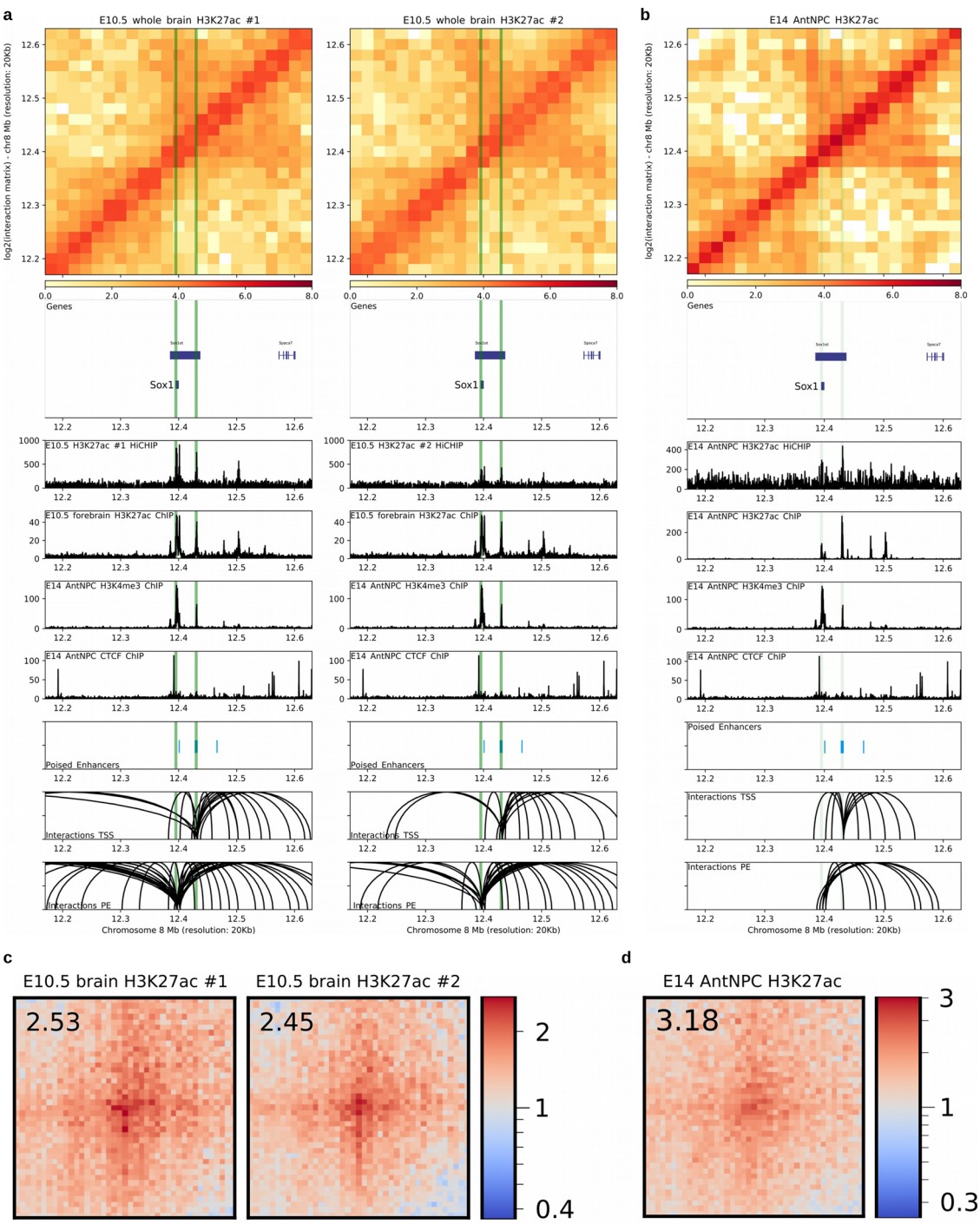

**Fig. 5 Interactions between PEs and their target genes are maintained once PE becomes active in either AntNPC or the embryonic mouse brain. a, b** H3K27ac HiChIP and ChIP-seq profiles generated in E10.5 mouse brains (**a**; two biological replicates) and AntNPC (**b**; both replicates merged), respectively, are shown around the *Sox1* locus. Both the *Lhx5* TSS and the PE *Sox1* (+35 kb) are marked with green vertical lines. Upper panel: heatmap of log2 interaction intensities based on H3K27ac HiChIP data generated in E10.5 mouse brain. Two medium panels: 1D HiChIP signals for H3K27ac in E10.5 mouse brain and H3K27ac, H3K4me3, and CTCF (for potential TAD boundaries) ChIP-seq profiles in AntNPC. Lower two panels: significant (*p* < 0.01 for E10.5 mouse brains; *p* < 0.05 for AntNPC; FitHiChIP) interactions were called using the H3K27ac HiChIP PETs (paired-end tags) and loops in which one of the anchors overlaps (±10kb) either the PE *Sox1* (+35 kb) or the *Lhx5* TSS are shown. **c, d** Significant (mESC H3K27me3 HiChIP data; *p* < 0.01; FitHiChIP) interactions between distal PEs (>10 kb from TSS) and ESC bivalent promoters (*n* = 526) were visualized as pileup plots using H3K27ac HiChIP data generated in H3K27ac HiChIP data generated in E10.5 mouse brain (**c**) or AntNPC (**d**). HiChIP interaction matrices were coverage-normalized (unbalanced). HiChIP pairwise interactions are shown 250 kb up- and downstream of each PE and bivalent promoter pair. The numbers in the upper left corners correspond to "loopiness" values (center intensity normalized to the intensity in the corners).

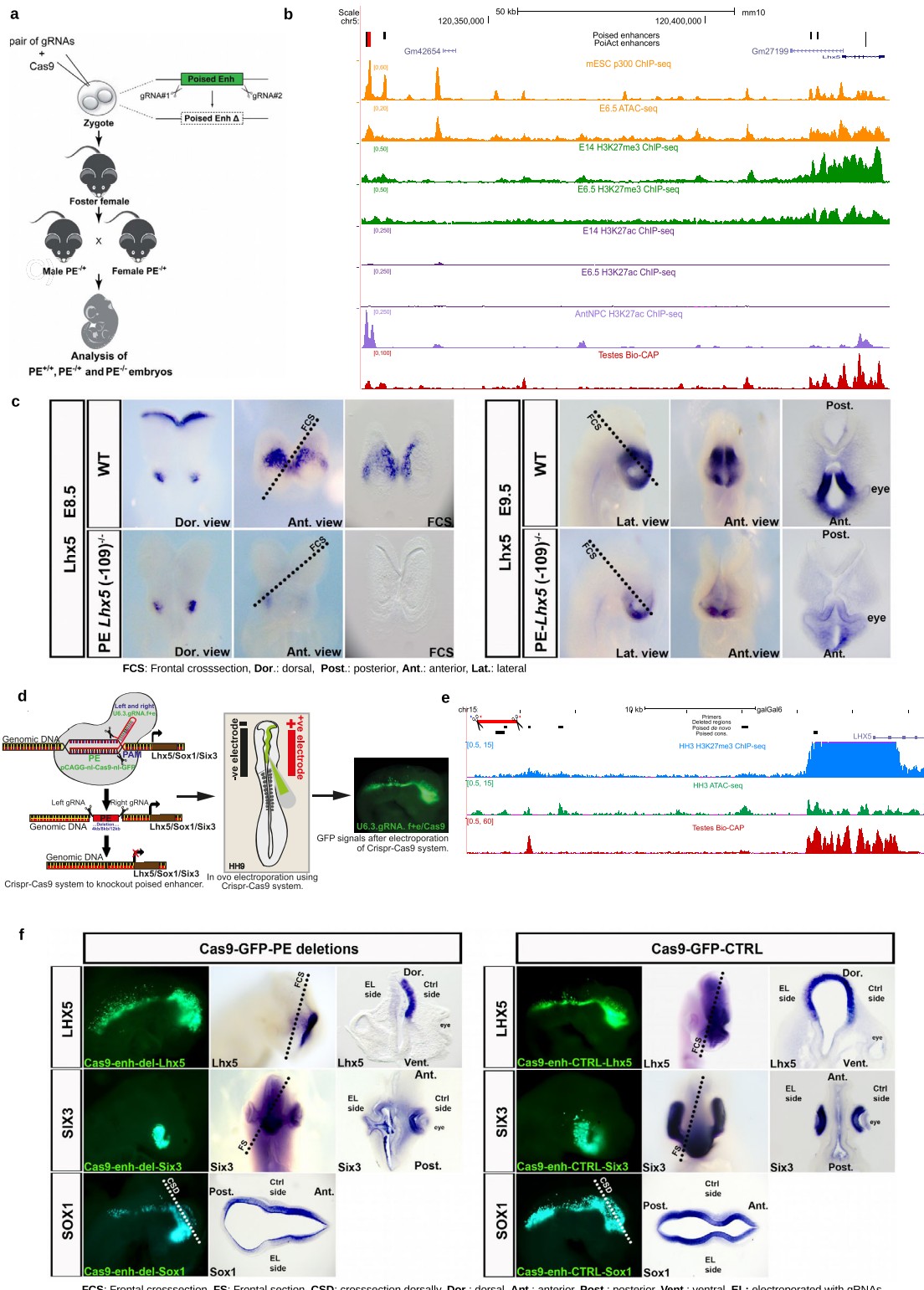

FCS: Frontal crosssection, Dor.: dorsal, Post.: posterior, Ant.: anterior, Lat.: lateral

FCS: Frontal crosssection, FS: Frontal section, CSD: crosssection dorsally, Dor.: dorsal, Ant.: anterior, Post.: posterior, Vent.: ventral, EL: electroporated with gRNAs

of major developmental genes within large regulatory domains[57–59]. Interestingly, although CGIs are considered as a vertebrate-specific genetic feature, sequences with equivalent tethering and regulatory functions might also exist in invertebrates, where they can also be important for the long-range induction of major developmental genes[60–63].

As mentioned above, the orphan CGI associated with PEs act as tethering elements physically linking these distal regulatory elements with their target genes. Mechanistically, in undifferentiated ESC, this tethering function seems to be mediated by PcG complexes recruited to the CGI present both at PEs and their target gene promoters[5,9,23,64]. Here we show that these PE–gene

**Fig. 6 Functional relevance of PEs during in vivo brain development. a** Graphical overview of the CRISPR/Cas9 experimental strategy used to generate mouse embryos with homozygous deletions of the PE *Lhx5* (−109kb). **b** ATAC-seq and ChIP-seq (p300, H3K27me3, and H3K27ac) profiles generated in mESC and E6.5 mouse epiblast are shown around the *Lhx5* locus. In addition, the Bio-CAP profiles generated in mouse testes are shown to illustrate the location of CGIs in the mouse genome, as well as H3K27ac ChIP-seq profiles generated in AntNPC to illustrate the activation of PE *Lhx5* (−109 kb) in AntNPC. The genomic region deleted in mouse embryos that includes the PE *Lhx5* (−109 kb) is highlighted in red. **c** RNA in situ hybridizations were performed to visualize *Lhx5* expression in WT and PE *Lhx5* (−109 kb)$^{-/-}$ mouse embryos at embryonic stages E8.5 (top rows) and E9.5 (bottom rows). Five PE *Lhx5* (−109 kb)$^{-/-}$ and four WT mouse embryos were analyzed for each developmental stage, with all PE *Lhx5* (−109 kb)$^{-/-}$ embryos showing the reduced *Lhx5* expression in the brain as compared to WT. **d** Graphical overview of the CRISPR/Cas9-based approach used to delete three genetically conserved PEs located within the *Lhx5*, *Sox1*, and *Six3* loci (left panel). Schematic diagram of in ovo electroporation technique in chick embryo at HH9. Mosaic knockout chicken embryos were generated by unilateral co-electroporation of CAGG > nls-Cas9-nls-GFP and U6.3>PE *Lhx5/Sox1/Six3*gRNAf + e vectors into the brain region (medium panel). GFP expression in brain and neural tube of an embryo electroporated with the Cas9 and gRNA-GFP vectors (right panel). **e** ATAC-seq and H3K27me3 ChIP-seq profiles generated in HH3 chicken epiblast are shown around the *Lhx5* locus. In addition, Bio-CAP profiles generated in chicken testes are also shown to illustrate the location of CGIs in the chicken genome. The genomic region deleted in chicken embryos that includes the mouse PE *Lhx5* (−109 kb) conserved in chicken is highlighted in red. **f** RNA in situ hybridizations were performed to visualize *Lhx5*, *Six3* or *Sox1* expression in HH14-HH16 chicken embryos that were unilaterally electroporated with (i) Left panels: Cas9-GFP and gRNAs flanking the conserved PEs associated with *Lhx5*, *Six3* and *Sox1* (Cas9-GFP-PE deletions); (ii) Right panels: Cas9-GFP and scrambled gRNA (Cas9-GFP-CTRL). For *Six3*, 16 Cas9-GFP-PE deletion embryos were analyzed and all of them showed decreased *Six3* expression and an aberrant eye phenotype on the electroporated side. For *Lhx5*, 20 Cas9-GFP-PE deletion embryos were analyzed and 14/20 embryos showed decreased *Lhx5* expression on the electroporated side, while 6/20 embryos showed mild or no reduction. For *Sox1*, 13 Cas9-GFP-PE deletion embryos were analyzed and 10/13 chicken embryos showed decreased *Sox1* expression on the electroporated side, while 3/13 embryos showed no reduction. For each of the previous three genes, we used six Cas9-GFP-CTRL chicken embryos as controls and no changes in gene expression were observed on the electroporated sides.

contacts are globally dependent on PRC1, while PRC2 seems to preferentially contribute to the interactions occurring within specific loci[5]. These results are in agreement with previous reports indicating that long-range interactions between PcG-loci are mediated by cPRC1 subunits[35,45–49,65], with PRC2 having a lesser and indirect contribution through its capacity to recruit cPRC1 to its genomic targets[66]. CGI can serve as recruitment platforms for other proteins containing CXXC domains (e.g. TET1, CFP1, MLL2/KMT2B), which are frequently part of important chromatin regulatory complexes (e.g. Trithorax (TrxG))[67–69]. It was recently shown that MLL2/KMT2B, an important component of TrxG complexes, can also contribute to the 3D chromatin organization of bivalent genes in pluripotent cells[51,70]. Interestingly, here we found that MLL2/KMT2B also facilitates the interaction between PE and their bivalent target genes. Therefore, PcG and TrxG complexes might cooperate rather than antagonize each other in pluripotent cells in order to mediate homotypic chromatin interactions within PE loci that facilitate future gene induction[39,48,71,72]. In addition to PcG and TrxG complexes, 3D chromatin organization is largely dependent on the combined effects of Cohesin and CTCF, which are necessary for the formation of TADs and other large regulatory domains through a loop extrusion mechanism[53,73–75]. Although PEs are not directly bound by either Cohesin or CTCF in ESC[5], we found that PE–gene contacts were diminished when either of these two architectural proteins were degraded. Therefore, loop extrusion might also facilitate the physical interactions between PEs and genes located within the same TAD. This is in contrast to the role of Cohesin as a negative regulator of inter-TAD interactions between PcG-bound genes[52].

Transcriptional and phenotypic robustness during development is believed to require complex regulatory landscapes whereby multiple enhancers redundantly control the expression of major cell identity genes[76–78]. In contrast, using ESC as an in vitro differentiation system, we previously showed that PEs can control the induction of genes involved in early brain development in a hierarchical and non-redundant manner[5]. Importantly, using both mouse and chicken embryos as experimental models, we have now confirmed the essential role of PEs for the proper induction of developmental genes during vertebrate embryogenesis. We propose that the privileged regulatory properties of PEs depend, at least partly, on nearby oCGI, which confer these

regulatory elements with unique chromatin and topological features[9]. However, it is important to mention that it is still unclear, both in vitro as well as in vivo, whether the "poised" state is actually important for enhancer function. Therefore, the oCGI might confer PEs their privileged regulatory properties once they become active in differentiating cells. Lastly, it is certainly possible that not all PEs acquire their unique chromatin and topological features already in pluripotent cells and this might occur later in lineage-restricted multipotent progenitors[79]. Therefore, future work should elucidate the full repertoire of PEs and interrogate their function in different somatic lineages and spatiotemporal contexts.

## Methods

**Cell culture**. 10 cm plates were coated with 0.1% gelatin generally overnight. Cells (E14 WT, EED−/−[41] and RING1a−/−RING1b$^{fl/fl}$[42] mESC) were thawed and resuspended in 10 ml of medium (serum + LIF). Standard serum + LIF medium contained 500 mL Knockout DMEM (Gibco 10829-018), 95 mL of filtered ES FBS (Gibco 16141-061), 5.9 mL of antibiotics (Hyclone SV30079.01), 5.9 mL Glutamax (Gibco 35050-038), 5.9 mL MEM NEAA (Gibco 11140-035), 4.7 mL titrated LIF (Miltenyi Biotec 130-095-777), and 1.3 mL Beta-mercaptoethanol 55 mM (Gibco 21985-023). Cells were split once (1/3 or 1/4) every two days. Ring1a−/−Ring1b$^{fl/fl}$ mESC were treated with Tamoxifen (OHT; 1 mM) for 72 h right after one passage and RING1B loss was confirmed by PCR genotyping and Western Blot.

**AntNPC differentiation**. E14 WT ESC grown in serum + LIF were treated with 2 mL TrypLE Express (Life Technologies). Cells were centrifuged (160 rcf) and then resuspended in 5 mL of N2B27 with 0.1% of BSA (Life Technologies) to get a single cell suspension. Next, cells were counted with the BioRad TC20 cell counter and 15,000 cells/cm$^2$ were plated in 10 mL of N2B27 with 0.1% of BSA and 10 ng/mL of bFgf (PeproTech, 100-18B). Cells were differentiated for five days and media was changed every day without any PBS washings in between: Day1 (10 mL of N2B27 with 0.004% of BSA and 10 ng/mL bFgf).; Day2 (10 mL of N2B27 with 0.004% of BSA, 10 ng/mL bFgf and 5 μM Xav939/Wnt inhibitor (Sigma-Aldrich, x3004-5mg)); Day3 (10 mL of N2B27 with 0.004% of BSA and 5 μM Xav939/Wnt inhibitor); Day4 (10 mL of N2B27 with 0.004% of BSA and 5 μM Xav939/Wnt inhibitor). On Day 5, cells were washed 1–2 with PBS, and collected for downstream analyses. Differentiation was assessed using RT-qPCR on a Light Cycler 480II comparing AntNPC d5 vs. E14 WT d0 relative gene expression levels (using the 2ΔCt method) with housekeeping gene (*Eef1a1*), pluripotency markers (*Pou5f1/Oct4* and *Nanog*), mesoderm marker (*T*) and ectoderm markers (*Six3* and *Lhx5*). Standard deviations were calculated from technical triplicate reactions and were represented as error bars.

**ATAC-seq**. Embryonic tissues were extracted and resuspended into single cells. ATAC-seq was essentially conducted following the protocol from Buenrostro et al.[80]. In short, single cells were centrifuged at 5000*g* for 5 min and supernatant

was removed. Cells were then lysed with 100 μl cold lysis buffer (10 mM Tris-HCl, pH 7.4, 10 mM NaCl, 3 mM MgCl$_2$, 0,1% IGEPAL CA-630) supplemented with 4 μl of protease inhibitor (1 tablet/2 mL conc.) for at least 15 min on ice. Immediately after lysis, nuclei were centrifuged at 6000$g$ for 10 min at 4 °C. The resulting pellet was resuspended in transposase reaction mix (25 μl 2x TD buffer, 10 μl transposase (Illumina), 15 μl nuclease-free H$_2$O) and incubated for 30 min at 37 °C. Finally, the sample was purified using Qiagen MinElute PCR purification kit according to the manufacturer's protocol.

**ChIP.** Cells were crosslinked in 1% of formaldehyde for 10 min (rotating at RT) with subsequent quenching by glycine (0.125 M; rotating at RT). Cells were washed twice with PBS and 1x protease inhibitor (04693159001, Roche), then flash frozen in liquid nitrogen and stored at −80 °C. Afterwards cells were thawed for ~30 min and lysed in 50 mM Hepes, 140 mM NaCl, 1 mM EDTA, 10% glycerol, 0.5% NP-40 and 0.25% TX-100 together with protease inhibitor (Lysis Buffer 1) for 10 min at 4 °C while rotating. After centrifugation (5 min, 2000 rcf, 4 °C), the supernatant was discarded and the pellet resuspended in 10 mM Tris, 200 mM NaCl, 1 mM EDTA, 0.5 mM EGTA, protease inhibitor (Lysis Buffer 2) and lysed for 10 min at 4 °C while rotating. After centrifugation (5 min, 2000 rcf, 4 °C), the supernatant was discarded and the pellet was resuspended in 10 mM Tris, 100 mM NaCl, 1 mM EDTA, 0.5 mM EGTA, 0.1% Na-Deoxycholate and 0.5% N-lauroylsarcosine with protease inhibitor (Lysis Buffer 3). Chromatin was then sonicated with ActiveMotif Sonicatior (amplitude: 25%, on=20 s, off=30 s, 20 cycles). The sonicated chromatin was centrifugated for 10 min, 16,000 rcf at 4 °C. The supernatant was collected and supplemented with 10% Triton X-100. 10% of the sonicated chromatin was kept as input DNA. For each ChIP reaction, 5 μg antibody were added to the remaining sonicated chromatin. ChIP samples were rotated vertically at 4 °C overnight (12–16 h) to bind antibody to chromatin. On the next day, 50–75 μl magnetic Dynabeads (Protein G) were washed three times (3X) in 1 mL cold Block Solution (0.5% BSA (w/v), 1x PBS). Antibody-bound chromatin was added to beads and inverted to mix. Then rotated vertically at 4 °C for at least 4 h. Afterwards, bound beads were washed 5X in 1 mL cold RIPA buffer (50 mM Hepes, 500 mM LiCl, 1 mM EDTA, 1% NP-40, 0.7% Na-Deoxycholate). Then, samples were washed once in 1 mL TE + 50 mM NaCl on ice and centrifuged for 3 min at 1000 rcf, 4 °C to remove all remaining TE. 210 μl Elution Buffer was added to the beads and DNA was eluted for 15 min in a thermoblock at 65 °C with shaking (900 RPM). Samples were centrifuged (1 min, 16,000 rcf, RT) and the supernatants were transferred (~200 μL) to fresh microfuge tubes. Both ChIP and input samples were then reverse-crosslinked and treated with RNAse and Proteinase K. Finally, DNA was extracted by phenol-chloroform followed by ethanol precipitation and resuspension in water. DNA content was measured using Qubit and the HS DNA Kit (Invitrogen Q32851). All antibodies used in this study have been previously reported as suitable for ChIP: H3K4me3 (39159, Active Motif), H3K27ac (39133, Active Motif), CTCF (61312, Active Motif), H3K27me3 (39155, Active Motif).

**HiChIP.** HiChIPs were performed as described[33] with some modifications. We generally replaced the ChIP protocol with the one described above, we cut and ligated overnight (NEB T4 Ligase, #M0202 instead of Invitrogen T4, 15224-041) and DNA was extracted with phenol-chloroform. For low cell numbers, we increased the centrifugation time to 30 min and 15 min after lysis, as well as 30 min after ligation to see a pellet more accurately. Generally, 12 cycles were used for Tn5 Nextera PCR amplification (Illumina Nextera DNA UD Indexes Kit). We aimed for 100 M read pairs for each run on a HiSeq 2500 sequencer (Illumina), except for the AntNPC samples which were sequenced at 50 M read depth. We used approx. 1/4 of a 10 cm plate for each cell culture HiChIP replicate. For each murine E10.5 brain replicate, we used at least 1.5 M cells.

**Generation of CRISPR/Cas9-mediated PE deletions in mice.** The CRISPR/Cas9 endonuclease-mediated PE deletions were generated by the CECAD in vivo Research Facility (ivRF) by pronuclear injection of the Cas9 nuclease mRNA and protein, tracrRNA, and crRNAs into C57BL/6 zygotes[81]. Cas9 nuclease (Addgene #1074181), tracrRNA (Addgene #1072532), and custom crRNA sequences were purchased from Integrated DNA Technologies (IDT; Coralville, Iowa, USA). The animals were housed and bred under standard conditions in the CECAD ivRF under a 12 h light cycle, at a temperature of 22 ± 2 °C, 55 ± 5% relative humidity and with food and water ad libitum. The breedings described were approved by the Landesamt für Natur, Umwelt, und Verbraucherschutz Nordrhein-Westfalen (LANUV), Germany (animal application 84-02.04.2015.A405).

**Designing and molecular cloning of chicken gRNA.** The template sequence flanking the PE region for each gene of interest (Lhx5, Sox1, and Sox3) was obtained from the UCSC Genome Browser and used for gRNA design. The gRNA sequences were designed using Benchling (Supplementary Data 1). We followed the standard principles to avoid off-target effects when choosing between multiple gRNA targets for each PE and cloned into the modified U6.3>gRNA.f + e backbone (Addgene #99139) as described in[55]. After cloning the gRNA into the backbone, the positive bacterial clones were identified using colony PCR with the corresponding forward and reverse gRNA oligo described in Supplementary Data 1. The resulting PCR products were analyzed by Sanger sequencing (SeqLab).

We used a control gRNA with a protospacer sequence not found in the chicken genome (GCAC-TGCTACGATCTACACC) which is already cloned into U6.3>Control.gRNA f + e vector, also provided by Prof. Dr. Marianne Bronner (Addgene #99140). To validate the CRIPSR/Cas9 targeting efficiency, PCR-based genotyping was done.

**Genotyping of PE deletions.** Whole chicken embryos were electroporated with pCAGG>nls-hCas9-nls-GFP together with either U6.3>PE_Lhx5 gRNA f + e, U6.3>PE_Sox1 gRNA f + e or U6.3>PE_Six3 gRNA f + e at stage HH9, then incubated until stage HH14-16 was reached. Live embryos were dissected in sterile PBS (1x) on ice and electroporated GFP-positive regions were isolated using surgical scissors. After isolation, neural tube and eye sections were pooled in a 1.5 ml tube separately for each embryo and used immediately for genotyping with one volume of Lysis Buffer (LyB; 50 mM KCL, 10 mM TRIS pH 8.3, 2.5 mM MgCl$_2$, 0.45% NP40 and 0.45% Tween 20), containing Proteinase K (1 μl of 20 μg/μl of Proteinase K for every 25 μl of LyB) and incubated at 55 °C for 1 h with frequent shaking to mix. The lysate was then heated to 95 °C for another 10 min to inactivate Proteinase K. We then tested for the presence of the PE deletions by PCR-based genotyping using the specific primers described in Supplementary Data 1.

To detect the PE Lhx5 deletion in mice, genomic DNA was isolated from ear punches and analyzed by PCR using the primers shown in Supplementary Data 1. The deletion was further confirmed by Sanger sequencing of the deletion-specific PCR products (Seqlab).

**Chicken embryos.** Fertilized chicken eggs (white leghorn; Gallus gallus domesticus) were obtained from a local breeder (LSL Rhein-Main) and incubated at 37 °C and 80% humidity in a normal poultry egg incubator (Typenreihe Thermo-de-Lux). Following microsurgical procedures, the eggs were re-incubated until the embryos reached the desired developmental stages. The developmental progress was determined according to the staging system of HH[82].

According to the relevant German legislation ("Tierschutz-Versuchsverordnung"), work with non-mammalian vertebrate embryos (e.g. chicken embryos) before 2/3 of their total developmental window does not require ethical approval. In all performed experiments, chicken embryos were kept no longer than stage HH14-16 (2-2.5 days), which is considerably earlier than 2/3 of the chicken embryo whole developmental window (20–21 days).

**In ovo electroporation.** Electroporations were performed using stage HH9 chicken embryos. 3.5–4 mL of albumin were removed by using a medical syringe to lower the blastoderm and make the embryo accessible for manipulation. The eggs were windowed, and the extra embryonic membrane was partially removed in the region to be electroporated. For knockout experiments, 5 μg/μl pCAGG>nls-hCas9-nls-GFP (Addgene #99141) and 3 μg/μl U6.3 > Lhx5/Sox1/Six3 gRNA f + e was microinjected together with the Fast Green solution (Sigma) at a 2:1 ratio to ease the detection of the injection site of the developing neural tube and eye respectively with the help of borosilicate glass capillaries and electroporated by placing the electrodes on each side of the microinjected neural tube/eye, and five square pulses of 80 V within 20 ms were applied to each embryo using the Intracel TSS20 OVODYNE Electroporator[83–85]. Control embryos were similarly electroporated with 1.5 μg/μl U6.3>Control.gRNA f + e along with pCAGG>nls-hCas9-nls-GFP. Following electroporation, the eggs were sealed with medical tape and re-incubated until the desired developmental stages (HH14-16) were reached.

**In situ hybridization (ISH).** At the desired stages, embryos were dispatched and fixed in 4% PFA/PBT overnight for ISH. Whole mount ISH of electroporated chicken embryos (HH14-HH16) and mutant/wt mice embryos (E8.5 and E9.5) was performed with probes against the target genes as described in[85,86]. For Lhx5, Sox1 and Six3, T7 promoter-containing PCR products were synthesized from stage HH9-HH14 chick cDNA. The gel-purified PCR products were used as templates for synthesis of antisense RNA probes using T7 polymerase enzyme. Primers are described in Supplementary Data 1. Riboprobes were labeled with a digoxigenin RNA labeling kit (Thermofisher, AM1324). Furthermore, mutant/wt mice embryos were tested for Lhx5, using DIG-labeled RNA probe against mouse Lhx5 (986 bp) (Supplementary Data 1) by cloning the template into the pCR™II-TOPO (Thermofischer) vector, digestion with XhoI and in vitro transcription using the SP6 polymerase.

**Chicken embryo sectioning and microscopy.** Selected embryos were sectioned using a vibratome (Leica) at a thickness of 30–35 μm. Light microscopy images were taken on a Olympus SZX16 stereomicroscope, To prepare the permanent slide, sections were embedded in Aquatex (Merck).

**ChIP- & ATAC-seq analysis.** Essentially the same analysis workflows were used for ChIP-seq and ATAC-seq data. Sequencing reads were mapped with bwa-0.7.7 mem[87], then converted from SAM to BAM (with samtools-1.2[88]) and sorted (with samtools-1.2). We then removed duplicate reads (with picard-tools-2.5.0[89]) and generated an index file (with samtools-1.2). Finally, bigWig/bedGraph files were

generated with deepTools[90] bamCoverage 2.5.7, normalized to 1x depth of coverage (reads per genome coverage), profiled and uploaded into the UCSC browser[91] (hub hosted on cyverse[92]). Peaks are called with macs 2.1.1.20160309[93]. The narrow peak mode and the corresponding default threshold ($q = 0.05$) were used to call ATAC-seq and p300 peaks. The broad peak mode and the corresponding default threshold ($q = 0.1$) were used to call H3K4me1, H3K4me3, H3K27ac, and H3K27me3 peaks. The ChIP-seq data from mouse WT in vitro pluripotent cell types (i.e. serum+LIF ESC, 2i ESC, and EpiLC) was obtained from Cruz-Molina et al.[5] and Bleckwehl et al.[94]. Mapping statistics for our generated data are listed in Supplementary Data 1. All publically available ChIP-seq and ATAC-seq data sets used in this work, including those generated by ENCODE[95], are listed in Supplementary Data 1. Multiple files of the same entity were merged using bigWigMerge.

**Enhancer calling**. The previous ATAC-seq and ChIP-seq peaks were used to call active, primed and poised enhancers using basic operations (INTERSECT, SUB-TRACT, MERGE) available in bedtools2-2.19.0[96].

Mouse in vitro pluripotent cell types: Poised and active enhancers in serum +LIF ESC were previously reported in Cruz-Molina et al.[5]. For the previous enhancer sets, those located proximal to gene TSS (±5 kb) were filtered out. H3K27ac, H3K4me1, and H3K27me3 broad peaks (default: $q \leq 0.1$) in 2i ESC and EpiLC were additionally filtered by requiring fold-enrichments over input of at least 5, 2, and 2, respectively. H3K27ac peaks were similarly identified in serum +LIF ESC. The resulting peaks were extended ±1 kb. p300 and ATAC-seq narrow peaks (default: $q \leq 0.05$) in 2i ESC and EpiLC were additionally filtered by requiring fold-enrichments over input of at least 4.

Poised enhancers—2i ESC and EpiLC ATAC-seq peaks overlapping (INTERSECT) with genomic regions enriched in both H3K27me3 and H3K4me1 in the corresponding cell type were identified and combined with the serum+LIF PEs via UNION. Those genomic regions located proximal to gene TSS (±5 kb) were filtered out. Then, genomic regions being enriched in H3K27ac in any of the in vitro pluripotent cell types were SUBTRACTED (H3K27ac peaks identified in 2i, serum+LIF, and EpiLC were combined via UNION. Finally, the resulting genomic regions were MERGED to define a total of 4191 unique PEs in in vitro mouse pluripotent cells.

Active enhancers—2i ESC and EpiLC ATAC-seq peaks overlapping (INTERSECT) with genomic regions enriched in both H3K27ac and H3K4me1 in the corresponding cell type were identified and combined with the serum+LIF active enhancers via UNION. Those genomic regions located proximal to gene TSS (±5 kb) were filtered out. Then, genomic regions being enriched in H3K27me3 in any of the in vitro pluripotent cell types were SUBTRACTED (H3K27me3 peaks identified in 2i, serum+LIF, and EpiLC were combined via UNION). Finally, the resulting genomic regions were MERGED to define a total of 14803 unique active enhancers in in vitro mouse pluripotent cells.

Primed enhancers—H3K4me1 peaks identified in 2i ESC and EpiLC were combined with the serum+LIF primed enhancers via UNION. Those genomic regions located proximal to gene TSS (±5 kb) were filtered out. Then, genomic regions being enriched in H3K27me3 or H3K27ac in any of the in vitro pluripotent cell types were SUBTRACTED (H3K27me3 peaks identified in 2i, serum+LIF, and EpiLC were combined via UNION; H3K27ac peaks identified in 2i, serum+LIF and EpiLC were combined via UNION. Finally, the resulting genomic regions were MERGED to define a total of 55812 primed enhancers in in vitro mouse pluripotent cells.

To avoid redundancies between the different enhancer groups, enhancers overlapping between each of the three previous categories (poised, active, and primed) were filtered out, except for those overlapping active and primed enhancers ($n = 855$), which were attributed to the active enhancer category.

PoiAct enhancers—PEs identified in the in vitro pluripotent cell types were intersected with either in vitro AntNPC H3K27ac peaks ($q \leq 0.1$; broad; fold-enrichment ≥ 5; extension ±1 kb) or in vivo E10.5 brain (fore-, mid- and hindbrain)[95] H3K27ac peaks ($q \leq 0.1$; broad; fold-enrichment ≥ 1; in vivo). To call H3K27ac peaks in E10.5 fore-, mid- and hindbrain, we overlapped (INTERSECT) the peaks identified in each of the two biological replicates available for each brain part (Supplementary Data 1).

Mouse E6.5 epiblast—to call in vivo mouse PEs, E6.5 epiblast ATAC-seq peaks (FC ≥ 5; $q = 0.05$; narrow peak mode) overlapping (INTERSECT) with genomic regions enriched in H3K27me3 (FC ≥ 2; $p \leq 0.01$; extension ±1 kb; broad peak mode) in the E6.5 epiblast were identified. Then, genomic regions enriched in H3K27ac in either E6.5 epiblast (FC ≥ 2; $p \leq 0.01$; broad peak mode) or in vitro pluripotent cell types (UNION of H3K27ac peaks identified in 2i ESC, serum+LIF ESC, EpiLC) were subtracted. Finally, genomic regions located proximal to gene TSS (±5 kb) were filtered out to define a total of 3057 PEs in the mouse E6.5 epiblast.

H3K4me1 was not used to define in vivo mouse PEs as ChIP-seq data for this histone mark was not available in the E6.5 epiblast. Moreover, the peak calling criteria for H3K27ac and H3K27me3 in the E6.5 epiblast were more relaxed than in the in vitro pluripotent cell types due to the overall lower quality of in vivo ChIP-seq data sets. To ensure that the identified in vivo PEs are not active in pluripotent cells and due to the lower quality of the in vivo ChIP-seq data, we subtracted H3K27ac regions identified in any of the investigated in vitro pluripotent cell types.

Human, chicken, and zebrafish—for human, chicken, and zebrafish, de novo PEs were called using similar criteria to those described for mouse cells, but we

extended each H3K27me3 and H3K27ac (H3K27ac only available for human ESC) peak by ±2.5 kb and we only considered regions located distally (>10 kb) from gene TSS. For calling ChIP-seq and ATAC-seq peaks we always used $q \leq 0.1$. Moreover, for the p300/ATAC-seq peaks we required the following fold-enrichment over input thresholds: human ≥ 3, chicken ≥ 3 and zebrafish ≥ 5. For the H3K27me3 peaks we required the following fold-enrichment over input thresholds: human ≥ 3, chicken ≥ 1, and zebrafish ≥ 3. The H3K27ac peaks in human ESC were required a fold-enrichment over input threshold ≥3.

All generated enhancer bed files are listed in Supplementary Data 2.

PE sets were annotated using GREAT 4.0.4[97] (human and mouse) and ConsensusPathDB Release 34 (15.01.2019)[98] (chicken and zebrafish). The used assemblies and annotations are listed in Supplementary Data 1.

**Conservation analysis**. All available UCSC vertebrates were considered if they had an available liftOver chain file from mm10 (Supplementary Data 1 for all used genome builds). Specific genome build versions were used to match Ensembl TSS annotations of the species in which Bio-CAP data was available[22]. Identity thresholds of 50% were used.

All species depictions are royalty-free/limited licensed under the stock IDs 57341071, 1666821223, 750090523, 99158564, 182837423, 194504681 & 1536755681.

**Non-methylated islands (NMI)**. NMI and Bio-CAP data were obtained from Long et al.[22]. Processed NMI bed files were initially used to calculate NMI to PE distances. Raw reads were downloaded from GEO[99] (GSE43512), aligned using bowtie2-2.2.0[100] and profiled using deepTools bamCoverage 2.5.7. Narrow NMI peaks were called using macs 2.1.1.20160309 callpeak ($q \leq 0.01$; "-extsize 300 -mfold 10 30").

**HiChIP**. Except for murine E10.5 brain samples, biological/technical replicates were merged. Reads were aligned and quality-controlled through HiC-Pro 2.10[101]. Initial anchor calling was conducted using macs 2.1.1.20160309 callpeak on bowtie alignments with default $q \leq 0.1$ (E14 mESC H3K27me3; E10.5 brain H3K27ac) or $q \leq 0.01$ (residual). The resulting peaks were passed to FitHiChIP 7.0[102] (29th April 2019) for final loop calling (in L_COV mode). Basic loop quality control statistics including distance plots were conducted using diffloop 1.14.0[103]. hichipper 0.7[104] was used to generate bedgraph files and ChIP signal profiles. BigWig files were generated after clipping bedGraph files using bedClip. Final tracks were visualized in WashU[105] or UCSC genome browsers. Loop anchors were overlapped with TSS (±7.5 kb) and annotated using ConsensusPathDB[98]. Hi-C matrices were processed and normalized using cooler 0.8.7[106] (5 kb resolution), then plotted with coolpup.py 0.9.2[107] using default settings with "-padding 100". HiChIP samples were coverage-normalized (unbalanced) and Hi-C samples were KR-balanced (balanced). Loopiness values were calculated on the single center pixel normalized to corners ("-enrichment 1 -norm_corners 1"). Single loci were visualized using HiCPlotter 0.8.1[108] with standard RefSeq annotations. PcG domains for looping analyses were called by overlapping significant peaks (macs2; narrow; $q = 0.1$) of EED[109] and RING1b[110] ChIP-seq (three replicates each against input; peaks intersected) binding sites. TAD evaluations were conducted using coordinates[111] included within the HiCPlotter package lifted over from mm9 to mm10. All residual statistical analyses were conducted using custom-made scripts in R (3.6.0; 2019-04-26).

**RNA-seq and scRNA-seq**. The expression of targets genes (evaluated through sign. E14 H3K27me3 HiChIP interactions) for different poised and PoiAct enhancer sets was plotted for AntNPC and mESC RNA-seq[5], as well as for blastulation, gastrulation, and neurulation samples from two scRNA-Seq data sets[112,113]. Expression differences between differentiation or developmental stages were evaluated using two-sided Wilcoxon tests. Processed data were obtained from the respective GEO entries. To circumvent batch effects in both plots, expression was divided by housekeeping genes (all available eukaryotic translation elongation factors and actin molecules) mean expression.

## Data availability

The data that support this study are available from the corresponding authors upon reasonable request. The ChIP-seq, ATAC-seq and HiChIP data generated in this study has been deposited in the GEO repository database under accession code GSE160657. The generated enhancer sets in this study are provided in Supplementary Data 2.

All public data sets used are listed here and in Supplementary Data 1: GSE155089, GSE65583, GSE89211, GSE41267, GSE69919, GSE41923, GSE74617, GSE73952, GSE43512, GSE124342, GSE100597, GSE98671, GSE87038, GSE38066, GSE24447, E-MTAB-7816, GSE23716, GSE66390, GSE125318, GSE76505, E-MTAB-6165, GSE76687, GSE70355. The source data are provided with this paper.

## Code availability

All analysis was conducted using established software wrapped in custom-written scripts. Scripts are available upon request.

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

## Acknowledgements

The authors thank the Rada-Iglesias lab members for insightful comments and critical reading of the manuscript, Elisabeth Kirst and Janine Altmüller (Cologne Center for Genomics; University of Cologne (UoC)) for technical assistance with next-generation sequencing, and the Regional Computing Center of the UoC (RRZK) for providing computing time on the DFG-funded High-Performance Computing (HPC) system CHEOPS, as well as support. We acknowledge and appreciate the CECAD in vivo Research Facility (Branko Zevnik) for the generation and maintenance of the deleted PE *Lhx5* mouse line. We also thank Anton Wutz and Miguel Vidal for generously providing us with the EED−/− and RING1a−/−*RING1b^{fl/fl}* mESC lines. Work in the Rada-Iglesias laboratory was supported by CMMC intramural funding (Germany), the German Research Foundation (DFG) (Research Grant RA 2547/1-3), "*Programa STAR-Santander Universidades, Campus Cantabria Internacional de la convocatoria CEI 2015 de Campus de Excelencia Internacional*" (Spain), the Spanish Ministry of Science, Innovation and Universities (Research Grant PGC2018-095301-B-I00) and the European Research Council (ERC CoG *"PoisedLogic"*; 862022). Giuliano Crispatzu is supported by funding within the CRU329 (DFG 386793560). Rizwan Rehimi is supported by funding within the SFB829 (DFG 73111208).

## Author contributions

G.C., R.R.R., A.R.I. conducted most research and prepared the manuscript. G.C. conducted bioinformatical analysis and HiChIPs. R.R.R. deleted PEs in HH9 chicken, did in situ and HH3 chicken ChIP−/ATAC-seqs. C.X. and H.B. deleted PEs in E8.5 and E9.5 mice. HB and RRR did in situ in mice deletions. G.C. and TP did PcG-null ChIPs and molecular evaluations. T.B. cultured 2i cells and differentiated EpiLC with

subsequent ChIP-seqs. S.C.M. technically assisted with initial HiChIP experiments. E.M. provided support with in vivo mouse experiments.

## Funding

## Competing interests

The authors declare no competing interests.
