## [Peer Review File · Nature Communications]

REVIEWER COMMENTS

Reviewer #1 (Remarks to the Author):

Crispatzu and Rehimy et al. investigated the conservation of pluripotency-associated poised enhancers (PEs) *in vivo* in terms of chromatin features, 3D genome interaction and functionality. The study of PE conservation *in vivo* has important biological implications. By generating and mining various types of genomic data, the authors characterized genetic and epigenetic features of PEs in mouse pluripotent cells and studied the conservation of these PEs during development and across different species. Furthermore, using genome editing approaches, the authors showed that conserved PEs are critical for the expression of major developmental genes *in vivo* in mouse and chicken embryos. However, there are issues with the data analysis, which may lead to misinterpretation of the results as listed below:

1) The authors should clearly define PEs, and other enhancers types such as primed and active enhancers. Currently, the distinction between primed and poised enhancers are not specified in the main text, and even in the method section, the definition is confusing and possibly flawed. It appears that the authors are defining primed enhancers based on "the presence of H3K4me1, and no H3K27ac or H3K27me3", and Poised enhancers based on "the presence of H3K27me3 and ATAC-seq/p300, and no H3K27ac, and active enhancers based on "the presence of H3K27acAC-seq/p300, and no H3K27me3". There are two main problems here. First, H3K4me1 is a general histone mark for enhancers (Calo & Wysocka, 2013; Creighton et al., 2010), and therefore should be used consistently when calling poised, active and primed enhancers. Second, because these enhancer types are defined using different genomic datasets and criteria, it's unclear whether they represent completely distinct groups or overlapping groups.

2) The advantage of defining enhancers in three *in vitro* conditions (mESC in serum+LIF, mESCs in 2i and EpiLC) is that one may compare these different conditions to study potential changes that correlate with different pluripotent or developmental stages. However, instead of calling enhancers separately in each condition, the authors pooled the data together. This approach not only defeats the purpose of getting data from three pluripotent conditions, but can also lead to incorrect characterization of enhancers (and enhancer types). For instance, enhancers that only exist in one or two of the three pluripotent cell populations may be missed by subtracting pooled histone modification ChIP-seq peaks.

3) This study aimed to determine if "PEs are functionally conserved *in vivo*". However, they only defined PEs based on *in vitro* data. To properly address the question, the authors need to also directly define PEs based on *in vivo* data.

4) All the ChIP-seq and ATAC-seq data should be properly normalized (e.g. by FPKM or RPKM). Normalizing the data by genome coverage can be problematic, because even using the same experimental conditions the genome coverage can vary depends on the cell type or state. Furthermore, when comparing data from different species (e.g. Fig. 1d & 2a), the signals should be standardized and the inherent differences (e.g. global histone modification level, genome size) should be considered when doing so to make sure that the results are comparable.

4) There is no QC result or quantification information for any of the sequencing experiments. It is, therefore, difficult to assess the quality of those experiments.

5) While the deletion studies (Fig 6) are nice and support the function of enhancers *in vivo*, they don't necessarily confirm the function of PEs. In other words, while these enhancers were "poised" at one point, they became active when gene expression was examined. The deletion study shows the requirement of active enhancers for gene expression, but doesn't necessary prove the requirement of the "poised" state.

Additional points.

- 1) Many figures should be improved for clarity and simplicity. Fig 1a is meant to show how PEs are defined, but it's confusing and should be revised. Fig 1e can be plotted into one graph. Most of the contents in Fig 2 can go to supplementary, because the major conclusion for this figure is redundant to what is shown in Fig 1. Many figures are poorly labeled. There are often cases where the labels are too small for the readers to see (e.g. Fig 6d, e). Fig. 3f & g were mislabeled.
- 2) The authors should be careful when interpreting correlation data, and some arguments need to be revised to be more accurate. For example, "some PEs remain acetylated in postnatal mouse brain tissues (Fig. S1c), suggesting that some of these regulatory elements might contribute not only to the induction but also to the maintenance of gene expression". It is certainly possible, but there is no further proof to establish the relationship between gene expression and the level of H3K27ac at PEs to support such an argument. Similarly, there is no causal relationship between the H3K27me3 signal at PEs and the distance from these PEs to nearby CGI to support the argument "the weaker H3K27me3 enrichment observed at conserved PEs in zebrafish embryos (Fig. 1d) could be explained, at least partly, by the frequent absence of nearby oCGI".
- 3) Please provide heatmaps for ChIP-seq and ATAC-seq profiles shown in main and supplementary figures.
- 4) For peak calling, the default q for MACS2 should be 0.05 instead of 0.1 as stated in the methods section. If the authors used q=0.1 for peak calling, they may be overcalling.
- 5) I do not see the relationship between Fig. 2C and the correlated statement made in the manuscript.
- 6) The manuscript did not mention how the mESC used for H3K27me3 HiChIP was treated (which condition?).
- 7) HiChIP results are all presented in 20kb resolution in heatmaps, which are blurry and not informative. If the matrices were processed at 5kb resolution as stated in the Methods, why do not show the heatmaps in higher resolution?
- 8) For Fig. 3c, how many of these promoters are bivalent?
- 9) Please also show interaction numbers called in each condition in each of the pile-up figures. please explain why using 'unbalanced' (Fig. 4b, 5c,d) in some cases and 'balanced' in other cases?
- 10) For Fig 4a, please explain what 'PET counts' are. Also, how many loops were called in each condition?
- 11) For Fig 6b, ChIP-seq signals using the same antibody should be scaled in the same way.

Reviewer #2 (Remarks to the Author):

This is an interesting manuscript that explores the functional properties of poised enhancers (PE) in vivo, which nicely complements a previous study by the same group on an in vitro differentiation system (<https://www.biorxiv.org/content/10.1101/2020.08.05.237768v1>). This class of enhancers, composed by TF binding sites coupled to a distal orphan CpG island that recruits Polycomb complexes, appears to play a crucial role in the activation of distant developmental genes also bound by Polycomb. In this previous study, the group demonstrates that orphan CpG islands in PEs are instrumental in bringing together enhancers and gene promoters, prior to their activation.

Here the authors bring these previous findings, initially limited to a handful of loci, to the genome-wide level. First, they identify putative PE as highly-enriched p300/ATAC and H3K27me3 loci in several vertebrate species, by combining publicly-available and newly-generated datasets. Interestingly, they observe that the PE epigenetic signature is conserved across all studied vertebrates and therefore was likely present in the last common ancestor of bony fishes. Then, using H3K27me3 HiChIP experiments in mESC they show that PE interact primarily with bivalent promoters located within the same TAD. Then, in order to shed light on

how such contacts are formed, they combine publicly-available HiC datasets and their own H3K4me3 HiChIP experiments on loss of function models of key chromatin organizers. By doing so, they could demonstrate that PE-promoter interactions are diminished in mutants of both the Polycomb (PRC1 particularly) and Trithorax complexes. Remarkably, interactions are also affected by the loss of TADs in CTCF and cohesin deignons, in stark contrast with previously described promoter-promoter interactions mediated by Polycomb. Finally, they demonstrate the conserved functional relevance of PE in vivo by the deletion of two of these elements (at the *Six3* and *Lhx5* loci), both in mouse and chicken. Such manipulations led to severe reduction of target gene expression and to developmental defects in the case of the *Six3* enhancer deletion.

Despite their biological relevance in developmental gene expression, PE are still understudied. Therefore, unraveling the functional properties of this class of regulatory elements will be of great interest for the field of gene regulation and appealing for the readership of Nature Communications. In addition, PE appear to be vertebrate-specific, which poses interesting questions regarding the evolution of these elements.

Following, we list our concerns regarding the manuscript:

Major comments

1. We are concerned with the quality of some HiChIP datasets, in particular with the ones using the H3K27ac antibody. According to the first tracks of Figure 5A-B, the libraries do not seem to be particularly enriched in H3K27ac peaks.

The authors should prove that the IP of the HiChIP experiments is comparable to the IP obtained in conventional ChIP-seq experiments. For instance, they could show heatmaps comparing the 1D HiChIP signal with the matching ChIP-seq signal around ChIP-seq peaks for every HiChIP experiment. Then, it would become clear which datasets have sufficient quality to be used to draw accurate conclusions.

For instance, the authors claim that PE-promoter contacts are also present in cell populations where the both PE and promoter are active. This is based on a H3K27ac HiChIP experiment where the enrichment for this histone mark is, to say the least, far from optimal. In order to sustain that claim, HiChIP experiments have to be improved.

2. On Figure 3C, the authors claim that PE interact mainly with bivalent promoters. Such conclusions might be true, but they definitely cannot be drawn from this analysis since a proper background control is lacking. The authors could, for instance, compare against the epigenetic signature of all H3K4me3 TSSs. Are those TSSs less enriched in H3K27me3 than those interacting with PE? If that is the case, then it would be fair to claim that PE contacts are enriched in bivalent promoters.

3. Across the manuscript, the authors claim that PE are enriched in the regulatory landscapes of neural genes. We find that the enrichment analysis does not necessarily support that claim, especially across species.

Figure 2C, which uses GREAT, is particularly confusing, since it shows expression enrichment for mouse, zebrafish and human (in the case of human using the expression of the mouse ortholog) and pathway enrichment for chicken. While in mouse neural terms might be predominant, that does not seem to hold for the rest of species. In fact, such results are not completely convergent with Figure 3D analysis, which seems to be cleaner, since it is focused on such promoters interacting with PE. There, pathways related to the development of the tree main germ layers appear (neural crest, hindbrain, endoderm, heart). My opinion is that the authors should focus instead in addressing if PE target genes tend to be important developmental regulators regardless of their expression domain. This seems to be already partially supported by the gene ontology terms shown in the same figure for mouse (pattern specification, DNA binding, and so on). It would be nice if that holds also for distant species such as zebrafish.

Minor comments

- 1- Figures 1 and 2 can be probably simplified and merged in a single figure.
 - 2- Figure 1A scheme is difficult to understand. I would recommend it to be carefully rethought and to include the species/models used and the experiments were performed/available for each one of them. The authors might consider adding the numbers for each category.
 - 3- Figure 2 caption title makes a distinction between "higher" and "lower" vertebrates which is discouraged in the evolutionary biology field and has a dubious relationship with actual phylogeny. Alternatively, it could be named as "Poised enhancers are a widespread feature across vertebrates" or "Poised enhancers are conserved between mammals and teleost fishes" or "Poised enhancers were present in the last common ancestor of bony fishes"(that include zebrafish and tetrapods).
 - 4- In general, the individual items of all figures can be redistributed and resized to avoid the presence of large blank spaces.
 - 5- The manuscript would benefit from discussing the relationship between the origin of PE and the origin of CGI (which are vertebrate-specific).
 - 6- In the section "Mouse PEs display high genetic and epigenetic conservation across mammals", on the 2nd paragraph there might be a typo.
- "..PEs in non-vertebrate species...": The authors might be referring to non-mammalian vertebrates.

Reviewer #3 (Remarks to the Author):

This is a very well written article that convincingly describes Poised Enhancers in vivo.

The work done is extensive and the problematic is approached from different angles. Additionally, the level of conservation of PE is explored across different taxa. The article definitely provides an important contribution to advance the field, not only in theoretical terms but also in methodological terms, since innovative methods are described to investigate PE.

I do have minor comments, mainly in relation to the clarification of some terms, but overall, I was very pleased to read this article:

Introduction, Line 14: Please explain briefly what an orphan CpG island is, and what is the difference with a canonic CpG island.

Results, lines 6-7: Please clarify what is meant by saying that PEs identified in different cells were 'pooled'. I assume that you are referring to merging of data, and if that is the case, please modify the text for clarity and exclude the term 'pool' because it is generally used to describe the mixing of biological (wet lab) samples from different sources.

Results, lines 3-5 of paragraph 4: It is good the authors mention that CpG islands are sometimes called non-methylated islands in the literature. In the remainder of the manuscript, however, I suggest the authors sticking to one of the terms, because sometimes CGI is used, sometimes NMI is used, and sometimes both are used. Please, be consistent with this term across the manuscript to avoid confusion.

Results, line 10 of paragraph 4: Change 'responsible, at least partly, to the unique....' for 'responsible, at least partly, for the unique....'

Results, paragraph 5: Please, clarify what is meant by a 'de novo PE'. One or two sentences to clarify the concept should suffice.

Discussion, 22nd line of the second paragraph: Change 'This is contrast to...' for 'This is in contrast to...'

Discussion, Last paragraph: This is more a question out of curiosity: you mentioned that PEs should be investigated more in depth in somatic lineages. What is the evidence for the importance of PEs in the germ line? Could PE be relevant there too?

Point-by-point response to the Reviewers.

“The chromatin, topological and regulatory properties of pluripotency-associated poised enhancers are conserved in vivo”

We would like to thank all the reviewers for their insightful and constructive comments, which have helped us to improve our work. Please, find below our responses to each of the concerns raised by the three reviewers.

REVIEWER COMMENTS

Reviewer #1 (Remarks to the Author):

Crispatzu and Rehimí et al. investigated the conservation of pluripotency-associated poised enhancers (PEs) *in vivo* in terms of chromatin features, 3D genome interaction and functionality. The study of PE conservation *in vivo* has important biological implications. By generating and mining various types of genomic data, the authors characterized genetic and epigenetic features of PEs in mouse pluripotent cells and studied the conservation of these PEs during development and across different species. Furthermore, using genome editing approaches, the authors showed that conserved PEs are critical for the expression of major developmental genes *in vivo* in mouse and chicken embryos. However, there are issues with the data analysis, which may lead to misinterpretation of the results as listed below:

We appreciate the reviewer’s insightful suggestions to improve our work.

1) The authors should clearly define PEs, and other enhancers types such as primed and active enhancers. Currently, the distinction between primed and poised enhancers are not specified in the main text, and even in the method section, the definition is confusing and possibly flawed. It appears that the authors are defining primed enhancers based on “the presence of H3K4me1, and no H3K27ac or H3K27me3”, and Poised enhancers based on “the presence of H3K27me3 and ATAC-seq/p300, and no H3K27ac, and active enhancers based on “the presence of H3K27acAC-seq/p300, and no H3K27me3”. There are two main problems here. First, H3K4me1 is a general histone mark for enhancers (Calo & Wysocka, 2013; Creighton et al., 2010), and therefore should be used consistently when calling poised, active and primed enhancers. Second, because these enhancer types are defined using different genomic datasets and criteria, it’s unclear whether they represent completely distinct groups or overlapping groups.

Firstly, in order to be consistent with our previous work, in serum+LIF ESC we used the active and poised enhancers described in Cruz-Molina et al. (Cruz-Molina et al., *Cell Stem Cell*, 2017). Then, following the reviewer’s suggestion, we have now included H3K4me1 to call poised, primed and active enhancers in both 2i ESC and EpiLC. Furthermore, to avoid redundancies between the different enhancer groups, enhancers overlapping between each of the three previous categories (poised, active and primed) were filtered out, except for those overlapping active and primed enhancers (n=855), which were only assigned to the active enhancer category.

The overall criteria used to define the poised, active and primed enhancers are now more extensively described in the Methods section (pages 19-20) as well as in the revised **Fig. 1A** and **Fig. S1A,B**, which we believe illustrate more clearly how the different enhancer groups were called. Moreover, we have also added a brief description of how the different enhancer types were called in the main text (page 3). Importantly, although the total number of enhancers belonging to each category has changed in the revised manuscript and, consequently, many analyses and figures have been modified accordingly, the main observations and conclusions of our work are still the same.

2) The advantage of defining enhancers in three *in vitro* conditions (mESC in serum+LIF, mESCs in 2i and EpiLC) is that one may compare these different conditions to study potential changes that correlate with different pluripotent or developmental stages. However, instead of calling enhancers separately in each condition, the authors pooled the data together. This approach not only defeats the purpose of getting data from three pluripotent conditions, but can also lead to incorrect characterization of enhancers (and enhancer types). For instance, enhancers that only exist in one or two of the three pluripotent cell populations may be missed by subtracting pooled histone modification ChIP-seq peaks.

Poised enhancers (i.e. ATAC-seq/p300 ChIP-seq peaks overlapping regions enriched in H3K27me3 and H3K4me1) are initially called in each condition separately. Then, we subtracted genomic regions enriched in H3K27ac in any of the *in vitro* pluripotent cell types (i.e. H3K27ac peaks identified in 2i+LIF, serum+LIF and EpiLC were combined via UNION). Finally, the resulting genomic regions were merged and potential TSSs filtered to define a total of 4191 unique PEs in *in vitro* mouse pluripotent cells. Therefore, PEs can exist in one or two pluripotent cell types as far as they are not enriched in H3K27ac in any of the three pluripotent states. The subtraction of H3K27ac regions present in any of the three pluripotent cell types was applied because our objective is to define PEs that might get activated upon pluripotent cell differentiation rather than during the transitions between pluripotent states.

As stated in the previous response, a more extensive description of the criteria used to call poised enhancers is presented in the **Methods** section (pages 19-20), as well as in the revised **Fig. 1A**. Furthermore, in the revised manuscript, we have also included heatmaps for p300/ATAC-seq, H3K27me3, H3K27ac and H3K4me1 signals (**Fig. S1C**) that show that the identified PEs display a rather similar chromatin signature in the three different pluripotent cell types.

3) This study aimed to determine if “PEs are functionally conserved *in vivo*”. However, they only defined PEs based on *in vitro* data. To properly address the question, the authors need to also directly define PEs based on *in vivo* data.

We agree with the reviewer in that it would be important to define mouse PEs *in vivo*. As described in the **Methods** section (page 20), we first identified E6.5 epiblast ATAC-seq peaks ($FC \geq 5$; $q = 0.05$; narrow peak mode) overlapping with genomic regions enriched in H3K27me3 ($FC \geq 2$; $p = 0.01$; extension +/-1kb; broad peak mode) in the E6.5 epiblast (*Zheng et al., Mol Cell, 2017*). Then, genomic regions enriched in H3K27ac in either the E6.5 epiblast ($FC \geq 2$; $p = 0.01$; extension +/-1kb; broad peak mode) or the *in vitro* pluripotent cell types (UNION of H3K27ac peaks identified in 2i+LIF ESC, serum+LIF ESC, EpiLC) were subtracted. Finally, genomic regions located proximal to gene TSS (+/- 5kb) were filtered out to define a total of 3057 PEs in the mouse E6.5 epiblast

(new **Fig. 1C**). Please note that H3K4me1 was not used to define *in vivo* PEs as ChIP-seq data for this histone mark is not available in the E6.5 epiblast. Moreover, the peak calling criteria for H3K27ac and H3K27me3 in the E6.5 epiblast were more relaxed than in the *in vitro* pluripotent cell types due to the overall lower quality of *in vivo* ChIP-seq data sets. To ensure that the identified PEs are not active in pluripotent cells and due to the lower quality of the *in vivo* ChIP-seq data, we subtracted H3K27ac regions identified in any of the investigated *in vitro* pluripotent cell types.

Out of those 3057 PE identified in the mouse E6.5 epiblast, 39.68% (1213/3057) overlap with our *in vitro* PE and 41.09% (1256/3057) are activated in the E10.5 brain (i.e. *in vivo* poiAct enhancers). On the other hand, 30.06% (1260/4191) of our *in vitro* PE were also called *in vivo*. Considering the limitations to generate high quality ChIP-seq data in the mouse epiblast, we believe that the previous overlaps are highly significant and further support the existence and relevance of PEs *in vivo*. These results are now described in the **Results** section (page 4) and **Fig. 1C**. Lastly, we would like to point out that in the previous manuscript version we already defined *in vivo* PEs in both chicken and zebrafish embryos (**Fig. 2A**).

4) All the ChIP-seq and ATAC-seq data should be properly normalized (e.g. by FPKM or RPKM). Normalizing the data by genome coverage can be problematic, because even using the same experimental conditions the genome coverage can vary depends on the cell type or state. Furthermore, when comparing data from different species (e.g. Fig. 1d & 2a), the signals should be standardized and the inherent differences (e.g. global histone modification level, genome size) should be considered when doing so to make sure that the results are comparable.

The ChIP-seq and ATAC-seq data in our study was normalized using *deepTools* (Ramirez *et al.*, *NAR*, 2014). RPKM is one of the *deepTools* recommendations for RNA-seq normalization to library size (i.e. so excessive long coding regions do not downgrade any other genes). However, for ChIP-seq/ATAC-seq, the RPGC (reads per genomic content (1x normalization)) normalization is recommended because it takes into consideration both the library size, as well as the effective genome size, which also involves considering repetitive and blacklisted regions (for more details: <https://deeptools.readthedocs.io/en/develop/content/tools/bamCoverage.html#Read%20coverage%20normalization%20options>). Therefore, we believe that the RPGC normalization is the most appropriate in order to compare different cell types and species.

4) There is no QC result or quantification information for any of the sequencing experiments. It is, therefore, difficult to assess the quality of those experiments.

QC and quantification metrics were calculated for all of our sequencing experiments using *picard-tools-2.5.0 CollectAlignmentSummaryMetrics* and are now included in **Table S1**.

5) While the deletion studies (Fig 6) are nice and support the function of enhancers *in vivo*, they don't necessarily confirm the function of PEs. In other words, while these enhancers were "poised" at one point, they became active when gene expression was examined. The deletion study shows the requirement of active enhancers for gene expression, but doesn't necessary prove the requirement of the "poised" state.

The reviewer is absolutely right and we actually think that showing whether the “poised” state is important or not for the regulatory function of these developmental enhancers is a major open question and one of the future objectives of our laboratory. However, the experiments to evaluate the importance of the “poised” state are not trivial and, thus, we think they are outside the current scope of our manuscript. Nevertheless, we acknowledge that it is important to mention that the presented work does not demonstrate whether the “poised” state is functionally relevant and the following sentences have been added to the revised manuscript:

- **Results** (page 10): *“However, whether the essential regulatory properties of these enhancers require a “poised” state previous to their full activation remains to be demonstrated.”*
- **Discussion** (page 12): *“However, it is important to mention that it is still unclear, both in vitro as well as in vivo, whether the “poised” state is actually important for enhancer function. Therefore, the oCGI might confer PEs their privileged regulatory properties once they become active in differentiating cells.”*

Additional points.

1) Many figures should be improved for clarity and simplicity. Fig 1a is meant to show how PEs are defined, but it’s confusing and should be revised. Fig 1e can be plotted into one graph. Most of the contents in Fig 2 can go to supplementary, because the major conclusion for this figure is redundant to what is shown in Fig 1. Many figures are poorly labeled. There are often cases where the labels are too small for the readers to see (e.g. Fig 6d, e). Fig. 3f & g were mislabeled.

Fig. 1A, as well as the related **Fig. S1A-B**, have been improved in order to better describe how the different enhancer groups, including PEs, are defined. **Fig. 1E** (now **Fig. 1F**) has been plotted into one graph. Font sizes in several figures, including **Fig. 6D,E** were increased and uninformative text removed in order to improve the overall clarity of the figures. **Fig. 3F,G** are now properly labeled.

Regarding **Fig. 2**, we decided not to move it to the supplementary material because, in contrast to **Fig. 1**, it is not based on enhancers identified through sequence conservation, but rather on PEs called *de novo* in the different vertebrate species. Therefore, in our opinion, **Fig. 2** provides complementary rather than redundant information to what is presented in **Fig. 1**. Lastly, the number of supplementary figures is already quite high and it might be better not to increase it any further based on the journal recommendations.

2) The authors should be careful when interpreting correlation data, and some arguments need to be revised to be more accurate. For example, “some PEs remain acetylated in postnatal mouse brain tissues (Fig. S1c), suggesting that some of these regulatory elements might contribute not only to the induction but also to the maintenance of gene expression”. It is certainly possible, but there is no further proof to establish the relationship between gene expression and the level of H3K27ac at PEs to support such an argument. Similarly, there is no causal relationship between the H3K27me3 signal at PEs and the distance from these PEs to nearby CGI to support the argument “the weaker H3K27me3 enrichment observed at conserved PEs in zebrafish embryos (Fig. 1d) could be explained, at least partly, by the frequent absence of nearby oCGI”.

We fully agree with the reviewer in that, without additional experimental evidences, the potential role of PEs in postnatal tissues is too speculative at this point. Therefore, the text (page 3) and figures (former **Fig. S1C**; **Fig. S2B** in the revised manuscript) in which H3K27ac levels in postnatal tissues were described have been removed in the revised manuscript.

On the other hand, our recent work in mESC (*Pachano et al., bioRxiv, 2020*; currently in second revision in *Nature Genetics*) clearly demonstrates that oCGI are necessary and sufficient for the recruitment of PcG complexes and H3K27me3 to nearby PEs. Moreover, we think that our statement regarding how the absence of oCGI might contribute to the low H3K27me3 enrichment of conserved PEs in zebrafish embryos is quite conservative: “...*might be explained, at least partly, ...*”. Therefore, in the revised manuscript we have slightly modified the previous statement, including a reference to our own recent work supporting the importance of oCGI for the H3K27me3 enrichment at PEs (page 5): “*Therefore, considering the important role of oCGI in mediating the recruitment of PcG to PEs in mESC⁹, the weaker H3K27me3 enrichment observed at conserved PEs in zebrafish embryos (Fig. 1E) might be explained, at least partly, by the frequent absence of nearby oCGI.*”

3) Please provide heatmaps for ChIP-seq and ATAC-seq profiles shown in main and supplementary figures.

We now provide heatmaps for the ChIP-seq and ATAC-seq profiles analyzed in serum+LIF ESC, 2i+LIF ESC and EpiLC (new **Fig. S1C**). In addition, in response to reviewer #2, we also provide heatmaps for the H3K27ac 1D HiChIP and ChIP-seq signals generated in E10.5 mouse brain and AntNPC (new **Fig. S5A,B**). Due to space limitations and to avoid increasing the number of supplementary figures, we do not think it is feasible to provide heatmaps for all the additional ChIP-seq and ATAC-seq data sets.

4) For peak calling, the default q for MACS2 should be 0.05 instead of 0.1 as stated in the methods section. If the authors used q=0.1 for peak calling, they may be overcalling.

As q=0.05 is the default q-value for narrow peak calling, we used this threshold now for p300 ChIP-seq and ATAC-seq data. However, since q=0.1 is the default cut-off for broad peak calling, this cut-off was maintained for the analysis of histone modifications in which peaks were identified using the broad peak calling mode in MACS2. This is described in detail in the **Methods** section (page 31-32).

5) I do not see the relationship between Fig. 2C and the correlated statement made in the manuscript.

In response to reviewer #2, the *in silico* annotation analysis presented in **Fig. 2C** have been modified and they are now focused on *Gene Ontology (GO) Biological Process* terms. These analysis suggest a general association of *de novo* PEs identified in different vertebrate species with developmental genes in general rather than with neural or neural crest related genes in particular. Therefore the previous statement in page 6 has been corrected accordingly: “*Furthermore, in all the*

investigated species, the de novo PE were strongly associated with genes involved in developmental processes, such as patterning and organogenesis”.

6) The manuscript did not mention how the mESC used for H3K27me3 HiChIP was treated (which condition?).

The mESC were grown under serum+LIF conditions. This has now been mentioned in the revised text (page 7).

7) HiChIP results are all presented in 20kb resolution in heatmaps, which are blurry and not informative. If the matrices were processed at 5kb resolution as stated in the Methods, why do not show the heatmaps in higher resolution?

The HiChIP matrices were processed at 5kb resolution in order to perform loop calling, which is facilitated by the focal interactions overlapping with significant "ChIP" peaks. A similar 5kb resolution is feasible to generate pile-up plots in which the HiChIP signals over hundreds of loci are averaged. However, the 5kb resolution is not appropriate to visualize HiChIP signals as heatmaps across individual loci given the depth at which our HiChIP libraries were sequenced (around 100M reads). In this regard, when Hi-C signals are visualized across individual loci, they are typically processed using 10-50kb bins despite the fact that Hi-C samples are typically sequenced at considerably higher depths.

8) For Fig. 3c, how many of these promoters are bivalent?

We have used the gene lists previously provided by Mikkelsen et al. (*Mikkelsen, T.S. et al., Nature, 2007*), in which a total of 17018 gene promoters were classified as either bivalent, H3K27me3-only, H3K4me3-only or unmarked based on their chromatin state in mESC. Out of these 17018 genes, we found that 1083 interacted with distal PEs in mESC. Notably, out of these 1083 PE-interacting genes (shown in **Fig. 3C**), 424 are bivalent ($n=424/2794$; $p < 2.2e-16$; $OR=2.63$; Fisher test), 40 H3K27me3-only ($n=40/160$; $p = 8.88e-14$; $OR=4.90$; Fisher test), 480 are H3K4me3-only ($n=480/9663$; $p = 2.64e-06$; $OR=0.77$; Fisher test) and 155 are unmarked ($n=155/4758$; $p < 2.2e-16$; $OR=0.495$; Fisher test). Therefore, bivalent and H3K27me3-only genes are significantly overrepresented among the genes interacting with PEs in mESC, while H3K4me3-only and unmarked genes are underrepresented. These overlaps and their significance (calculated using Fisher tests) are now presented in the **Results** section (page 7).

9) Please also show interaction numbers called in each condition in each of the pile-up figures. please explain why using ‘unbalanced’ (Fig. 4b, 5c,d) in some cases and ‘balanced’ in other cases?

Overall, HiChIP samples were coverage-normalized (unbalanced) and Hi-C samples were KR-balanced (balanced) and this is now mentioned in the **Methods** and in the corresponding figure legends. The *coolpup.py* package, which we used to normalize the HiChIP and Hi-C samples in order to generate pile-up plots recommends using the unbalanced normalization method when working with non-standard Hi-C methods such as HiChIP.

On the other hand, the number of contacts being plotted in the different pile-up figures has been added to the corresponding figure legends.

10) For Fig 4a, please explain what 'PET counts' are. Also, how many loops were called in each condition?

Number of loops are now included in **Fig. 4A**. Please note that the total number of called loops is significantly decreased in the $\text{Ring1a}^{-/-}\text{Ring1b}^{\text{fl/fl}}$ ESC, despite the fact that H3K4me3 signals (**Fig. S4C**) and the overall quality of the HiChIP libraries (**Table S1**) generated in this cell line are similar to those obtain in WT ESC. As now mentioned in page 8, this probably reflects the involvement of PRC1 in mediating not only interactions between PE and bivalent genes but also active enhancer-gene contacts (Loubiere et al., Science Advances, 2020).

On the other hand, the explanation of what PET counts means (PET = paired-end tags) has been included in the **Fig. 4A** legend.

11) For Fig 6b, ChIP-seq signals using the same antibody should be scaled in the same way.

This has now been corrected in the revised manuscript.

Reviewer #2 (Remarks to the Author):

This is an interesting manuscript that explores the functional properties of poised enhancers (PE) in vivo, which nicely complements a previous study by the same group on an in vitro differentiation system (<https://www.biorxiv.org/content/10.1101/2020.08.05.237768v1>). This class of enhancers, composed by TF binding sites coupled to a distal orphan CpG island that recruits Polycomb complexes, appears to play a crucial role in the activation of distant developmental genes also bound by Polycomb. In this previous study, the group demonstrates that orphan CpG islands in PEs are instrumental in bringing together enhancers and gene promoters, prior to their activation.

Here the authors bring these previous findings, initially limited to a handful of loci, to the genome-wide level. First, they identify putative PE as highly-enriched p300/ATAC and H3K27me3 loci in several vertebrate species, by combining publicly-available and newly-generated datasets. Interestingly, they observe that the PE epigenetic signature is conserved across all studied vertebrates and therefore was likely present in the last common ancestor of bony fishes. Then, using H3K27me3 HiChIP experiments in mESC they show that PE interact primarily with bivalent promoters located within the same TAD. Then, in order to shed light on how such contacts are formed, they combine publicly-available HiC datasets and their own H3K4me3 HiChIP experiments on loss of function models of key chromatin organizers. By doing so, they could demonstrate that PE-promoter interactions are diminished in mutants of both the Polycomb (PRC1 particularly) and Trithorax complexes. Remarkably, interactions are also affected by the loss of TADs in CTCF and cohesin deignons, in stark contrast with previously described promoter-promoter interactions mediated by Polycomb. Finally, they demonstrate the conserved functional relevance of PE in vivo by the deletion of two of these elements (at the *Six3* and *Lhx5* loci), both in mouse and chicken. Such manipulations led to severe reduction of target gene expression and to developmental defects in the case of the *Six3* enhancer deletion.

Despite their biological relevance in developmental gene expression, PE are still understudied. Therefore, unraveling the functional properties of this class of regulatory elements will be of great interest for the field of gene regulation and appealing for the readership of Nature Communications. In addition, PE appear to be vertebrate-specific, which poses interesting questions regarding the evolution of these elements.

We appreciate the overall positive impression of the reviewer about our work as well as the insightful and constructive suggestions to improve it, which we address below.

Following, we list our concerns regarding the manuscript:

Major comments

1. We are concerned with the quality of some HiChIP datasets, in particular with the ones using the H3K27ac antibody. According to the first tracks of Figure 5A-B, the libraries do not seem to be particularly enriched in H3K27ac peaks. The authors should prove that the IP of the HiChIP experiments is comparable to the IP obtained in conventional ChIP-seq experiments. For instance, they could show heatmaps comparing the 1D HiChIP signal with the matching ChIP-seq signal around ChIP-seq peaks for every HiChIP experiment. Then, it would become clear which datasets

have sufficient quality to be used to draw accurate conclusions. For instance, the authors claim that PE-promoter contacts are also present in cell populations where the both PE and promoter are active. This is based on a H3K27ac HiChIP experiment where the enrichment for this histone mark is, to say the least, far from optimal. In order to sustain that claim, HiChIP experiments have to be improved.

Firstly, we think that the negative perception that the reviewer got about our H3K27ac HiChIPs might be caused, at least partly, by our choice to select the *Lhx5* locus to illustrate those HiChIPs in **Fig. 5A,B**. *Lhx5* expression in the developing brain is positionally restricted, being preferentially expressed in the roof plate. Therefore, its promoter and associated enhancers are likely to be enriched in H3K27ac only in a fraction of the brain cells analyzed. As a result, both the 1D HiChIP and ChIP-seq profiles in E10.5 brain samples look noisier than at other loci showing more widespread expression within the brain (e.g. *Sox1* and *Six3*). Therefore, in the revised manuscript we are now showing the *Sox1* locus in **Fig. 5A,B** for both E10.5 brain and AntNPC, while the *Lhx5* and *Six3* loci are shown in **Fig. S5A** (for E10.5 brain).

More importantly, following the reviewer's suggestion, we have now generated heatmaps and average plot profiles in which the H3K27ac 1D HiChIP and ChIP-seq signals in either E10.5 brain (new **Fig. S6A**) or AntNPC (new **Fig. S6B**) are plotted around bivalent promoters (in ESC), as well as different enhancer groups: PEs that get activated in either E10.5 brain or AntNPC (poiAct), PE that do not get activated in either E10.5 brain or AntNPC (non-poiAct) and active enhancers (in ESC). Although the 1D HiChIP signals show higher background levels, it is also clear that the relative H3K27ac enrichments observed for the different enhancer groups is actually similar when considering either the 1D HiChIP or ChIP-seq signals, with poiAct and bivalent promoters being particularly enriched in E10.5 brain and poiAct enhancers in AntNPC, thus supporting the overall quality of the H3K27ac HiChIP samples. Furthermore, our computational pipeline (**Methods**, page 21) to call loops in the HiChIP data requires (i) that at least one of the loop anchors is enriched in H3K27ac according to MACS2 (e.g. $q \leq 0.1$ for H3K27ac HiChIP in E10.5 brain) and (ii) that the loop is considered statistically significant ($p \leq 0.05$) according to FitHiChIP. This ensures that the called H3K27ac HiChIP loops display focal H3K27ac enrichments over background for at least one of the loop anchors. Using this loop calling strategy, we identified 67748-94731 loops in each of the two E10.5 brain HiChIP replicates ($p < 0.05$) and 49552 loops in the AntNPC HiChIP ($p < 0.05$). Importantly, as now described in page 9, these H3K27ac HiChIP loops showed significant overlaps with the PE-bivalent gene contacts identified in mESC (Fig. 3B) both in E10.5 brain (50.76-54.56% overlap; +/-10kb anchor extension) and in AntNPC (36.69% overlap; +/-10kb anchor extension).

On the other hand, upon closer inspection of the 1D HiChIP signals for the H3K4me3 HiChIP sample generated in AntNPC (former **Fig. 5C**) we realized that it had poorer quality and we have removed it from the revised manuscript.

2. On Figure 3C, the authors claim that PE interact mainly with bivalent promoters. Such conclusions might be true, but they definitely cannot be drawn from this analysis since a proper background control is lacking. The authors could, for instance, compare against the epigenetic signature of all H3K4me3 TSSs. Are those TSSs less enriched in H3K27me3 than those interacting with PE? If that is the case, then it would be fair to claim that PE contacts are enriched in bivalent promoters.

In order to address this important point, we have used the gene lists previously provided by Mikkelsen et al. (*Mikkelsen, T.S. et al., Nature, 2007*), in which a total of 17018 gene promoters were classified as either bivalent, H3K27me3-only, H3K4me3-only or unmarked based on their chromatin state in mESC. Out of these 17018 genes, we found that 1083 interacted with distal PEs in mESC. Notably, out of these 1083 PE-interacting genes (shown in **Fig. 3C**), 424 are bivalent (n=424/2794; $p < 2.2e-16$; OR=2.63; Fisher test), 40 H3K27me3-only (n=40/160; $p = 8.88e-14$; OR=4.90; Fisher test), 480 are H3K4me3-only (n=480/9663; $p = 2.64e-06$; OR=0.77; Fisher test) and 155 are unmarked (n=155/4758; $p < 2.2e-16$; OR=0.495; Fisher test). Therefore, bivalent and H3K27me3-only genes are significantly overrepresented among the genes interacting with PEs in mESC, while H3K4me3-only and unmarked genes are underrepresented. These overlaps and their significance (calculated using Fisher tests) are now presented in the **Results** section (page 7).

3. Across the manuscript, the authors claim that PE are enriched in the regulatory landscapes of neural genes. We find that the enrichment analysis does not necessarily support that claim, especially across species. Figure 2C, which uses GREAT, is particularly confusing, since it shows expression enrichment for mouse, zebrafish and human (in the case of human using the expression of the mouse ortholog) and pathway enrichment for chicken. While in mouse neural terms might be predominant, that does not seem to hold for the rest of species. In fact, such results are not completely convergent with Figure 3D analysis, which seems to be cleaner, since it is focused on such promoters interacting with PE. There, pathways related to the development of the tree main germ layers appear (neural crest, hindbrain, endoderm, heart). My opinion is that the authors should focus instead in addressing if PE target genes tend to be important developmental regulators regardless of their expression domain. This seems to be already partially supported by the gene ontology terms shown in the same figure for mouse (pattern specification, DNA binding, and so on). It would be nice if that holds also for distant species such as zebrafish.

We would like to thank the reviewer for raising this important point. Following the reviewer's advice, in the revised **Fig. 2C** we have now performed the *in silico* annotation of the PEs using *Gene Ontology (GO) Biological Process* terms for all the considered species instead of using *Expression* terms as shown in the former **Fig. 2C**. These GO annotations were performed with either *GREAT* for mouse and human or with *ConsensusPathDB* for zebrafish and chicken. *ConsensusPathDB* was used for these two species as they are not supported by the latest version of *GREAT*. Importantly, these new analyses (**Fig. 2C**) show that, in accordance with the results shown in **Fig. 3D**, PEs are preferentially associated with major developmental regulators involved in processes such as patterning, morphogenesis and organogenesis and without an obvious bias towards the neural lineage. Therefore, the previous statements regarding a preferential association of PEs with neural genes have been eliminated across the revised manuscript, in which we have included the following sentences instead:

- page 6: “Furthermore, in all the investigated species, the *de novo* PE were strongly associated with genes involved in developmental processes, such as patterning and organogenesis (**Fig. 2C**; **Fig. S3E-F**)”.
- page 7: “In agreement with this significant overrepresentation of bivalent and H3K27me3-only genes, the genes interacting with PEs tend to be preferentially involved in developmental processes (e.g. patterning, morphogenesis) (**Fig. 3D**).”

Minor comments

1- Figures 1 and 2 can be probably simplified and merged in a single figure.

Although we appreciate the reviewer's suggestion, we have decided to keep **Fig. 1** and **Fig. 2** separate from each other as we think they provide complementary rather than redundant information: **Fig. 1** is focused on enhancers identified through sequence conservation, while **Fig. 2** shows analyses related to PEs called *de novo* in the different vertebrate species. We think that this is a better alternative than merging these two figures and moving some of the panels to the supplementary material, which already includes a high number of figures.

2- Figure 1A scheme is difficult to understand. I would recommend it to be carefully rethought and to include the species/models used and the experiments were performed/available for each one of them. The authors might consider adding the numbers for each category.

Fig. 1A, as well as the related **Fig. S1A-B**, have been extensively modified in order to better describe how the different enhancer groups, including PEs, are defined. Furthermore, as suggested by the reviewer, these figures now include the number of enhancers detected in the different pluripotent states as well as the overlaps between them.

3- Figure 2 caption title makes a distinction between “higher” and “lower” vertebrates which is discouraged in the evolutionary biology field and has a dubious relationship with actual phylogeny. Alternatively, it could be named as “Poised enhancers are a widespread feature across vertebrates” or “Poised enhancers are conserved between mammals and teleost fishes” or “Poised enhancers were present in the last common ancestor of bony fishes”(that include zebrafish and tetrapods).

We apologize for using those incorrect terms. The **Fig. 2** caption has been re-named following one of the reviewer's suggestions: “*Poised enhancers are a widespread feature across vertebrates*”.

4- In general, the individual items of all figures can be redistributed and resized to avoid the presence of large blank spaces.

We have changed the distribution and size of several figure panels in order to minimize blank spaces.

5- The manuscript would benefit from discussing the relationship between the origin of PE and the origin of CGI (which are vertebrate-specific).

This topic was briefly mentioned in the first paragraph of the Discussion section. We have now extended this paragraph as follows (page 11): “*We also showed that the main regulatory function of these orphan CGI is to serve as tethering elements that bring PEs and their CpG-rich target genes into physical proximity⁹. Furthermore, the oCGI might also contribute to the high sequence conservation of PEs by protecting them from CpG methylation⁹ and, thus, from accumulating C>T mutations. Therefore, we propose that the association of distal enhancers with CGI might represent an ancestral regulatory mechanism in vertebrate genomes that enables the precise and specific induction of major developmental genes within large regulatory domains⁵⁷⁻⁵⁹. Interestingly,*

although CGI are considered as a vertebrate-specific genetic feature, sequences with equivalent tethering and regulatory functions might also exist in invertebrates, where they can also be important for the long-range induction of major developmental genes⁶⁰⁻⁶³.

6-In the section “Mouse PEs display high genetic and epigenetic conservation across mammals”, on the 2nd paragraph there might be a typo. “.PEs in non-vertebrate species...”: The authors might be referring to non-mammalian vertebrates.

Yes, we were referring to non-mammalian vertebrates. The typo has been corrected.

Reviewer #3 (Remarks to the Author):

This is a very well written article that convincingly describes Poised Enhancers *in vivo*. The work done is extensive and the problematic is approached from different angles. Additionally, the level of conservation of PE is explored across different taxa. The article definitely provides an important contribution to advance the field, not only in theoretical terms but also in methodological terms, since innovative methods are described to investigate PE.

We truly appreciate the overall positive impression of the reviewer about our work.

I do have minor comments, mainly in relation to the clarification of some terms, but overall, I was very pleased to read this article:

Introduction, Line 14: Please explain briefly what an orphan CpG island is, and what is the difference with a canonical CpG island.

The term orphan CpG island was first used by Illingworth et al. (*Illingworth et al., PLoS Genet, 2010*) to identify those CpG islands that were not associated with annotated gene promoters. This explanation has been added to the **Introduction** in page 2.

Results, lines 6-7: Please clarify what is meant by saying that PEs identified in different cells were ‘pooled’. I assume that you are referring to merging of data, and if that is the case, please modify the text for clarity and exclude the term ‘pool’ because it is generally used to describe the mixing of biological (wet lab) samples from different sources.

We have now changed the description of how the PEs were identified in different cell types, avoiding to use the term “pooled”: (page 3) “Next, PEs identified in S+L mESC, 2i mESC and EpiLC were combined, resulting in a total of 4191 unique mouse PEs”. Moreover, we have extensively modified **Fig. 1A** in order to better illustrate how the PEs were actually defined in the investigated *in vitro* pluripotent cell types.

Results, lines 3-5 of paragraph 4: It is good the authors mention that CpG islands are sometimes called non-methylated islands in the literature. In the remainder of the manuscript, however, I suggest the authors sticking to one of the terms, because sometimes CGI is used, sometimes NMI is used, and sometimes both are used. Please, be consistent with this term across the manuscript to avoid confusion.

Following the reviewer’s advice, we have decided to consistently use the term CGI throughout the manuscript. Moreover, we have added the following sentence in order to explain this decision (page 5): “The CGI identified through Bio-CAP are typically referred to as NMI²². However, to avoid possible confusions, from now on we will simply use the term CGI regardless of whether these genetic features were identified based on their genetic composition or by Bio-CAP.”

Results, line 10 of paragraph 4: Change ‘responsible, at least partly, to the unique....’ for ‘responsible, at least partly, for the unique....’

The sentence has been corrected.

Results, paragraph 5: Please, clarify what is meant by a ‘de novo PE’. One or two sentences to clarify the concept should suffice.

The following sentence has been added in the results section (page 6): “*The term de novo is used to define PEs that are directly identified using epigenomic data generated in each of the investigated vertebrate species in contrast to those solely defined by sequence conservation (Fig. 1D-F).*”

Discussion, 22nd line of the second paragraph: Change ‘This is contrast to...’ for ‘This is in contrast to...’

The sentence has been corrected.

Discussion, Last paragraph: This is more a question out of curiosity: you mentioned that PEs should be investigated more in depth in somatic lineages. What is the evidence for the importance of PEs in the germ line? Could PE be relevant there too?

We have recently investigated enhancers during germline specification using an *in vitro* system in which ESC are differentiated into PGC-like cells (PGCLC) (Bleckwehl *et al.*, *bioRxiv*, 2020; <https://www.biorxiv.org/content/10.1101/2020.07.07.192427v1>). Based on the data presented in this recent pre-print, the majority of active enhancers in PGCLC do not show H3K27me3 enrichments in ESC. Nevertheless, based on a preliminary analysis, we determined that 9.21% (n=386/4191) of the PEs reported in our manuscript overlap H3K27ac peaks in PGCLC, including a prominent enhancer associated with *Prdm14*, one of the master PGC regulators. Therefore, it is possible that a few PEs might also play important regulatory functions in the germ line.

REVIEWERS' COMMENTS

Reviewer #1 (Remarks to the Author):

The authors did a great job revising the manuscript! I have to say I still find Figure 1A not super straightforward to understand. It would be great if the authors can improve the clarity of this figure if possible.

Reviewer #2 (Remarks to the Author):

The authors have successfully addressed all our concerns and we do not have further comments